



# Open cells can decrease the mixing of free-tropospheric biomass burning aerosol into the south-east Atlantic boundary layer

Steven J. Abel[1], Paul A. Barrett[1], Paquita Zuidema[2], Jianhao Zhang[2], Matt Christensen[3], Fanny Peers[4], Jonathan W. Taylor[5], Ian Crawford[5], Keith N. Bower[5], and Michael Flynn[5]

[1]Met Office, Fitzroy Road, Exeter, EX1 3PB, UK
[2]Rosenstiel School of Marine and Atmospheric Science, University of Miami, Miami, USA
[3]Atmospheric, Oceanic & Planetary Physics, Department of Physics, University of Oxford, Oxford, UK
[4]College of Engineering, Mathematics, and Physical Sciences, University of Exeter, Exeter, UK
[5]Centre for Atmospheric Science, School of Earth and Environmental Sciences, University of Manchester, Manchester, UK

**Correspondence:** Steven Abel (steven.abel@metoffice.gov.uk)

**Abstract.** This work presents synergistic satellite, airborne and surface based observations of a Pocket of Open Cells (POC) in the remote south-east Atlantic. The observations were obtained over and upwind of Ascension Island during the CLouds and Aerosol Radiative Impacts and Forcing (CLARIFY) and the Layered Smoke Interacting with Clouds (LASIC) field experiments. A novel aspect of this case-study is that an extensive free-tropospheric biomass burning aerosol plume that had been

transported from the African continent was observed to be in contact with the boundary layer inversion over the POC and the surrounding closed cellular cloud regime. The in-situ measurements show marked contrasts in the boundary layer thermodynamic structure, cloud properties, precipitation and aerosol conditions between the open cells and surrounding overcast cloud field.

The data demonstrate that the overlying biomass burning aerosol was mixing down into the boundary layer in the stra-
tocumulus cloud downwind of the POC, with elevated carbon monoxide, black carbon mass loadings and accumulation mode aerosol concentrations measured beneath the trade-wind inversion. The stratocumulus cloud in this region was moderately polluted and exhibited very little precipitation falling below cloud base. A rapid transition to actively precipitating cumulus clouds and detrained stratiform remnants in the form of thin quiescent veil clouds was observed across the boundary into and deep within the POC. The sub-cloud layer in the POC was much cleaner than that in the stratocumulus region. The clouds in
the POC formed within an ultra-clean layer (accumulation mode aerosol concentrations $\sim$few cm$^{-3}$) in the upper region of the boundary layer, that was likely to have been formed via efficient collision-coalescence and sedimentation processes. Enhanced Aitken mode aerosol concentrations were also observed intermittently in this ultra-clean layer, suggesting that new particle formation was taking place. Across the boundary layer inversion and immediately above the ultra-clean layer, accumulation mode aerosol concentrations were $\sim 1000$ cm$^{-3}$. Importantly, the airmass in the POC showed no evidence of elevated carbon
monoxide over and above typical background conditions at this location and time of year. As carbon monoxide is a good tracer for biomass burning aerosol that is not readily removed by cloud processing and precipitation, it demonstrates that the open cellular convection in the POC is not able to entrain large quantities of the free-tropospheric aerosol that was sitting directly on





top of the boundary layer inversion. This suggests that the structure of the mesoscale cellular convection may play an important role in regulating the transport of aerosol from the free-troposphere down into the marine boundary layer.

We then develop a climatology of open cellular cloud conditions in the south-east Atlantic from 19 years of September MODIS Terra imagery. This shows that the maxima in open cell frequency ($> 0.25$) occurs far offshore and in a region where

subsiding biomass burning aerosol plumes may often come into contact with the underlying boundary layer cloud. If the results from the observational case-study applied more broadly, then the apparent low susceptibility of open cells to free-tropospheric intrusions of additional cloud condensation nuclei could have some important consequences for aerosol-cloud interactions in the region.

## 1 Introduction

Huge quantities of atmospheric aerosol particles are generated from biomass burning in the African sub-continent every year (van der Werf et al. , 2010). Much of this aerosol is transported westwards in the free-troposphere over the south-east Atlantic Ocean between June and October, above one of the worlds largest semi-permanent stratocumulus cloud fields (Adebiyi et al. , 2015). As the biomass burning aerosols are partially absorbing, they can exert a net warming of the atmospheric column and lead to a reduction in the outgoing flux at the top of atmosphere when overlaying these highly reflective clouds (a positive direct

effect), although the sign and magnitude of the direct effect is very sensitive to the underlying cloud fraction (Chand et al. , 2009). The warming of the free-troposphere can also act to strengthen the boundary layer temperature inversion and therefore reduce the entrainment of dry free-tropospheric air into the cloud layer below, especially when the vertical separation between the biomass burning aerosol and cloud top is small (Herbert et al. , 2019). The result is often to increase the amount of cloud condensate and brighten the stratocumulus (a negative semi-direct effect) e.g. Johnson et al. (2004); Wilcox. (2010). As the

base of the free-tropospheric aerosol gradually descends due to large-scale subsidence in the region, it can begin to mix down into the marine boundary layer. Recent observational studies have shown evidence of biomass burning aerosol in the boundary layer far offshore at Ascension Island (Zuidema et al. , 2018) and in regions closer to the African coast (Diamond et al. , 2018). Once these aerosols have been entrained into the boundary layer they can provide an additional source of cloud condensation nuclei (CCN), that can then result in a modification of the cloud properties (Diamond et al. , 2018) and a brightening of the

cloud field (a negative indirect effect) e.g. Lu et al. (2018). The aerosols can also then warm the layer in which the clouds form and promote decoupling of the boundary layer, suppressing moisture transport from the sea-surface to the clouds and reduce the liquid water path (a positive semi-direct effect) (Johnson et al. , 2004; Hill and Dobbie , 2008; Zhang and Zuidema , 2019). This diverse and competing set of aerosol-cloud-radiation interactions has resulted in potentially large but poorly constrained effects of how biomass burning aerosols impact the climate system in the south-east Atlantic. For example, two of the most

recent state-of-the art modelling studies have both shown that the net effect of the aerosol perturbations are to cool the region by modulating the large-scale cloud field (Gordon et al. , 2018; Lu et al. , 2018). However, the largest contributors to this cooling differ between models. In the study of Gordon et al. (2018), cloud adjustments that increase liquid water path (LWP) and cloud fraction due to the stabilization of the boundary layer from aerosol induced free-tropospheric warming dominates,



whereas the cloud microphysical effect from additional CCN entrained into the boundary layer is the largest contributor in the Lu et al. (2018) simulations. One of the important controlling factors for both of these mechanisms depends on how and when the free-tropospheric aerosol plumes mix down into the boundary layer.

In this paper, we present an observational case-study examining if the underlying cloud mesoscale structure changes the efficiency of this net free-tropospheric to boundary layer flow of aerosols. We draw on a combination of synergistic measurements made during the CLouds and Aerosol Radiative Impacts and Forcing (CLARIFY) experiments deployment of the Facility for Airborne Atmospheric Measurements (FAAM) BAe-146 research aircraft to Ascension Island (7.9° S, 14.4° W) in August to September 2017 and detailed ground based observations from the Layered Smoke Interacting with Clouds (LASIC) deployment of the Atmospheric Radiation Measurement (ARM) User Facility on Ascension Island in 2016-2017 (Zuidema et al. , 2018). These experiments form part of a recent larger concerted international effort undertaken to obtain a set of comprehensive in-situ and remote sensing measurements in the south-east Atlantic that will further improve our understanding of the role that biomass burning aerosols play in modulating the climate system in the region (Zuidema et al. , 2016a).

The case-study examines a well developed 'Pocket of Open Cells' (POCs) that co-incided with a large free-tropospheric biomass burning aerosol plume overlaying both the POC and surrounding stratocumulus cloud regimes. The term 'Pocket of Open Cells' refers to the occurrence of open cellular convection embedded in broad regions of unbroken closed cell stratocumulus. They are readily observed in satellite imagery in the large sub-tropical stratocumulus cloud decks and form within regions of initially unbroken cloud e.g. Wood et al. (2008) and Fig. 1 of this paper. There is still a lot of uncertainty in what the key mechanisms are that drive POC formation events in nature, partly due to there being no in-situ measurements of these events to date. However, model studies do show that the initiation of precipitation and it's evaporation below cloud base can drive circulation changes that promote the transformation of an overcast cloud field into open cells e.g. Savic-Jovcic and Stevens (2018); Wang and Feingold (2009a, b). It is therefore conceivable, although not the subject of this study, that the mixing of biomass burning aerosol into the boundary layer in the south-east Atlantic could act to reduce the occurrence of POC formation by suppressing precipitation formation. Once formed however, these mesoscale features are typically sustained for several days as they persist along the boundary layer flow. There have been several in-situ measurements of these mature POC features in the south-east and north-east Pacific stratocumulus decks, that document the contrasting conditions in fully developed POCs and the surrounding cloud (e.g. Stevens et al. (2005); Petters et al. (2008); Wood et al. (2008, 2011); Terai et al. (2014)). These observations and the aforementioned model studies show that POCs are maintained by aerosol-cloud-precipitation feedbacks, with stark differences in the mesoscale organization, dynamics, and microphysics within the different cloud regimes. A common feature is that they all show that the mature POCs exhibit significant enhancements in sub-cloud precipitation and much lower concentrations of boundary layer accumulation mode aerosol particulates than in the surrounding stratocumulus. A question then arises as to how the cloud in a well-developed POC may respond to a large CCN perturbation from entrained biomass burning aerosols. The idealised model study of Wang and Feingold (2009b) does show that an abrupt change in the CCN concentration in a mature POC can shut off precipitation and lead to cloud fraction increases, although this is not sufficient for the open cells to fully transition to the closed cell state in that case. That said, even a moderate change in cloud fraction could have important consequences for the direct and indirect effects in the region. The model studies of Berner





et al. (2011, 2013) also suggest that entrainment may be much weaker in POCs compared to the surrounding stratocumulus cloud field, which could limit how readily overlying biomass burning aerosol can be entrained into a POC. A key focus of this work therefore examines if there is observational evidence of differences between how subsiding biomass burning aerosol plumes are mixed down into the measured POC and surrounding overcast cloud regimes.

The paper is structured as follows. Section 2 briefly describes the datasets used in this study. The case-study is then introduced in Sect. 3 and an overview of the flight patterns performed in Sect. 4. The in-situ airborne observations of aerosol, cloud and boundary layer structure made both downwind of the POC and within the POC are then presented in Sect. 5 to 8. The view of these conditions from surface based measurements on Ascension Island are then presented in Sect. 9. Section 10 then takes a broader view of open cell conditions in the south-east Atlantic in order to put the results of the case-study into context.

Finally, a summary and discussion is presented in Sect. 11.

## 2 Datasets

### 2.1 Aircraft instrumentation

The Facility for Airborne Atmospheric Measurements (FAAM) BAe-146 research aircraft was equipped with a comprehensive suite of instrumentation during the CLARIFY campaign in order to make measurements suitable for studying aerosol-radiation and aerosol-cloud interactions. This included remote sensing instrumentation and in-situ measurements of the aerosol physical,

chemical and optical properties, cloud microphysics, thermodynamics and trace-gas chemistry. The instruments pertinent to this study include the following. Cloud and precipitation particle size distributions (PSDs) were measured with a variety of wing-mounted cloud physics probes. These instruments measure the concentration of hydrometeors as a function of particle size (in discrete size bins). In brief, cloud droplets (approximately 2 to 52 $\mu$m diameter) were measured with a Cloud Droplet

Probe (CDP). The CDP was calibrated using a ten point bead calibration before each flight day. Precipitation sized drops were measured with a 2D-Stereo (2DS) probe (10 to 1280 $\mu$m diameter) and a Cloud Imaging Probe (CIP-100) probe (100 $\mu$m to 6.4 mm diameter). Data from these different instruments are combined to produce a 1 Hz composite PSD following the methodology of Abel and Boutle (2012). A bulk measure of the total condensed water content from a Nevzorov total water content (TWC) sensor is also used. This measurement represents the combined liquid water content (LWC) of cloud drops

and precipitation sized particles. The Nevzorov data are baselined following the method of Abel et al. (2014). The LWC from cloud drops only is estimated by integrating the CDP PSD. The number concentration of aerosol particles were measured with a Passive Cavity Aerosol Spectrometer Probe (PCASP) for sizes between $\sim$ 0.1 and 3 $\mu$m and with a TSI 3786 Condensation Particle Counter (CPC) for all aerosols larger than about 2.5 nm. Both the PCASP and CPC data are only examined out of cloud and precipitation. Refractory black carbon (BC) mass concentrations are derived from a SP2 instrument following the method

described in Taylor et al. (2014). We also utilize carbon monoxide (CO) measurements from an AERO AL5002 instrument. Vertical wind information is given by the aircraft turbulence probe, temperature from a loom sensor mounted in a non-deiced Rosemount housing and humidity from a WVSS2 hygrometer (Vance et al. , 2015). A downward pointing Leosphere lidar is used to measure cloud top height when the aircraft was flying above the boundary layer, following the method of Kealy et al.


(2017). The vertical integral of aerosol extinction measured with a cavity ringdown system at 405 and 658 nm wavelengths (Davies et al. , 2019) is also used to calculate above cloud aerosol optical depth (AOD).

## 2.2 Surface based observations

We use a variety of surface based measurements from the LASIC ARM site on Ascension Island (Zuidema et al. , 2018).
This includes radiosonde profiles to examine the boundary layer thermodynamic structure. Aerosols and chemistry data used include refractory BC mass concentrations measured with a SP2, carbon monoxide and cloud condensation nuclei (CCN) measurements. The CCN data presented are at a supersaturation of 0.46 ±0.005 %. The majority of condensation particles with diameters larger than 10 nm are activated at this supersaturation (Zuidema et al. , 2018). Cloud base height is examined from a ceilometer. Data from a vertically pointing cloud radar (Ka-band) is used to give a more detailed picture of the cloud and
precipitation structure above the surface site. We also derive a boundary layer cloud top height product from the radar. This is calculated by looking at the highest range gate in the boundary layer where the radar reflectivity exceeds a -25 dBZ threshold. In addition to the LASIC ARM data, the Version 3 Level 1.5 total column AOD from the AERONET site at Ascension Island airport is also used (Giles et al. , 2019). We also examine radiosonde data from St Helena, which is approximately 1300 km upstream of Ascension Island given the mean boundary layer wind direction.

## 2.3 Satellite data

A range of satellite imagery and data products are used to give some wider context to the in-situ measurements. This includes infra-red and true-color imagery from the Spinning Enhanced Visible and Infrared Imager (SEVIRI) and the Moderate Resolution Imaging Spectroradiometer (MODIS). MODIS cloud top effective radius ($r_{eff}$) imagery derived from the 3.7 $\mu$m wavelength channel is also used (Platnick et al. , 2017a). We choose to use the 3.7 $\mu$m retrieval as it should be less suscepti-
ble to artefacts that can arise in conditions where biomass burning aerosol overlays boundary layer clouds (Haywood et al. , 2004). We also examine data from a new algorithm that performs a joint retrieval of cloud optical properties and aerosol optical depth overlying clouds from SEVIRI (Peers et al. , 2019), which should not suffer from such artefacts. To examine the vertical profiles of aerosol and cloud, we look at snapshots of the vertical feature mask from the CALIPSO (version 4.2) and CATS (version 3.0) spaceborne lidars (Winker et al. , 2009; Yorks et al. , 2016). The MODIS liquid cloud fraction and the fine mode
aerosol AOD from the monthly averaged L3 atmosphere product data collection 6.1 (Levy et al. , 2013; Platnick et al. , 2017b) are also utilised.

## 2.4 Trajectory data

Backward and forward trajectories from measurement locations are calculated using the Met Office Numerical Atmospheric-dispersion Modelling Environment (NAME) model (Jones et al. , 2007). The driving meteorological data are taken from the
operational Met Office global NWP analysis (N1280 grid-spacing, 70 vertical levels and the GA6.1 science configuration (Walters et al. , 2017)).



## 3   The case-study

Figure 1 shows snapshots of SEVIRI 10.8 $\mu$m brightness temperature imagery at selected times throughout the 4th and 6th September 2017, to illustrate the temporal evolution of the POC feature. The location of Ascension Island is shown with a green star. The main POC feature that was measured by the aircraft is labeled 'A' and a secondary POC feature is labeled 'B'.

The 04 UTC image on the 4th September shows the emergence of the region of open cells in POC 'A' surrounded by overcast cloud. This feature rapidly grows in horizontal extent as it is advected with the boundary layer flow to the north-west in the next 24 hr. At 04 UTC on the 5th September, the secondary POC feature 'B' forms to the south-east of POC 'A'. Both POC features continue to advect along the boundary layer flow towards Ascension Island in the north-west of the imagery. At 10 UTC on the 5th September, POC 'A' and 'B' cover an area of approximately 180,000 and 65,000 km$^2$ respectively. POC 'A'

is still a well defined feature that is surrounded by overcast stratiform cloud on the morning of the 5th September, but a large-scale cloud clearance on the POC's northern edge erodes the stratiform cloud layer into the afternoon of the 5th September as shown in Fig 1e. At the same time, the satellite imagery shows that the south-eastern edge of POC 'A' has merged with the north-western edge of POC 'B'. The two aircraft flights described in this study were flown on the morning of the 5th September in the stratiform cloud to the north-west of the POC, and then on the afternoon of the 5th September within the POC and across

the POC boundary into the cloud-free conditions associated with the large-scale cloud clearance. Given that the boundary layer flow is south-easterly, we will use the terminology 'downwind' of the POC throughout this work to refer to the airmass where the aircraft made in-situ measurements to the north-west of POC 'A'. By 00 UTC on the 6th September, the southern edge of the remnants of POC 'A' advect over Ascension Island and the associated thermodynamic, aerosol and cloud conditions were measured at the LASIC ARM site. Also included in Fig 1 are three trajectories calculated using the NAME model. The

trajectories are initialized in the boundary layer at 500 m altitude, from the position and time of aircraft measurements made in the stratiform cloud downwind of the POC (red), from the aircraft measurements within the POC (blue) and as the POC edge moves over the LASIC site on Ascension Island (green). The stars are the start-points of the trajectories and the position along each trajectory at the times of the individual satellite images are indicated with an open circle. All three trajectories originate towards the south-east and advect north-westwards with the typical boundary layer flow in the region.

Satellite data are analysed to examine how the cloud fraction, cloud top effective radius and the above cloud aerosol optical depth vary along the boundary layer trajectories in both the POC and downwind stratiform cloud regions that are shown in Fig 1. The satellite data are averaged over a $1 \times 1$ degree latitude-longitude box around each trajectory. Fig 2a shows the SEVIRI liquid cloud fraction. Data along the downwind trajectory (red points) shows overcast cloud conditions from the 3rd September to around midday on the 5th September. The cloud fraction then drops off rapidly to about 10 %, due to the large-scale clearance

in the stratiform cloud layer downwind of the POC that is seen between the imagery in Fig. 1d and e. The two trajectories that follow the POC (blue and green) also initially have 100% cloud fraction on the 3rd September. The cloud fraction then begins to decrease in the early hours of the 4th September, in accordance with the timing of the POC formation shown in Fig 1a. As the POC feature develops along the trajectory and grows in size, the cloud fraction continues to decrease to about 50 % on the morning of the 5th September.





The day-time satellite cloud top effective radius from SEVIRI is shown in Fig 2b. Before the formation of the POC on the 3rd September, the effective radius is approximately 8 to 10 $\mu$m on all three trajectories. However, there are marked differences in the microphysical characteristics of the stratiform and POC cloud regions when the POC forms and these differences then persist along the trajectories towards Ascension Island. The trajectory that is representative of the downwind stratiform cloud

layer (red) maintains effective radius values of around 5 to 10 $\mu$m throughout the period. This is contrasted with the data along the POC trajectories (blue and green), which show a continual increase after the POC forms from the 4th to the 5th September. By the 5th September, the SEVIRI retrieval has values in excess of 30 $\mu$m in the POC. The much larger cloud drop sizes in the POC are indicative of a cloud region that is more conducive to forming precipitation than the overcast stratiform cloud downwind. Data from the aircraft measurements are also included (upward pointing triangles) and they are consistent with the

large differences seen in the satellite data. The purpose of this is not to provide a detailed evaluation of the satellite retrievals, but to illustrate that the differences seen in the satellite data between the POC and downwind cloud conditions are consistent with in-situ observations. The in-situ cloud microphysical data will be examined in more detail in Sect. 7.

The day-time above cloud AOD retrieval from SEVIRI along each trajectory is shown in Fig 2c. The retrieval suggests that absorbing biomass burning aerosols were present above the boundary layer throughout the period and across both cloud

regimes, with AOD values that typically ranged between 0.2 and 0.5. There is an indication that there is an increasing trend in above-cloud AOD along the trajectories as they move northwards, although the variability is large, particularly on the 5th September. Also included in Fig 2c are values calculated from aircraft profiles above the boundary layer in both cloud regimes using the cavity ring-down system and total column AOD values from the AERONET site on Ascension Island on the afternoon of the 5th September. Both the aircraft and AERONET data are interpolated to a wavelength of 550 nm from neighbouring

channels in order to match that used in the SEVIRI retrieval. As with the effective radius data, the purpose is to illustrate that the in-situ measurements also show aerosol loadings that are broadly consistent with the satellite data. The in-situ aerosol observations will be examined further in Sect. 5 and 6.

The satellite above cloud AOD retrievals from SEVIRI are unable to determine if the free-tropospheric biomass burning aerosol was in contact with the boundary layer cloud along the trajectories as the AOD is a basic integrated value of the aerosol

extinction only. Spaceborne lidar such as CALIPSO and CATS are able to provide vertical information on the location of the aerosol and cloud, but the data are very limited both spatially and temporally. In Fig. 3, CALIPSO and CATS data that pass near to back trajectories initialized at the location of aircraft measurements made in the POC are presented. The trajectories were started at altitudes of 500 m (black line) and 1.5 km (red line), that correspond to heights below cloud base and in the upper part of the boundary layer (see Fig. 5). The 500 m altitude trajectory is the same as the blue line in Fig. 2. Following this trajectory

back to the 3rd September before the POC formed, there is a CALIPSO overpass that tracked very close by (point C on Fig. 3a and d). The CALIPSO data shows an elevated aerosol plume between about 3 and 5 km altitude at the corresponding latitude. There is however a clear slot beneath the base of this aerosol layer and above the boundary layer cloud, which is located at an altitude of 1.5 km. This indicates that the overlying smoke was not mixing into the boundary layer on the 3rd September before the POC formed. Further north on the CALIPSO overpass early on the 3rd September (latitude $\sim$ 9 to 14° S), there is

evidence that the base of the smoke plume is in contact with the underlying cloud. Later, on the 4th September there was a





CATS overpass that crossed upstream of the 500 m boundary layer trajectory. This also shows evidence of overlying biomass burning aerosol in contact with the underlying cloud to the north of about 15° S and a gap between the elevated aerosol plume and boundary layer cloud to the south of this (Fig. 3c). In addition, Fig. 3 e shows a vertical profile of water vapour from a radiosonde released at St Helena Island (16° S, 6° W) at 11:15 UTC on the 4th September. This is approximately 250 km to the west of the 500 m trajectory shown in Fig. 3 a and would have also been to the west of the POC feature at this time (see Fig 1 a and b). The sounding shows structure in the water vapour mixing ratio in the free-troposphere between the boundary layer top at 1.3 km and 3.7 km, with values in excess of 2 g kg$^{-1}$. This is likely to be associated with the transport of air and biomass burning aerosol from the continent (Adebiyi et al. , 2015). Whilst not definitive evidence, both the CALIPSO and CATS data suggest that aerosol-cloud contact was not prevalent south of ∼ 15° S along the POC trajectory, although the St Helena sounding suggests that the aerosol base may have lowered to the west of the POC.

To further examine when the POC airmass may have come into contact with overlying biomass burning aerosol, Fig. 3 b shows the altitude of the trajectories and the thicker lines show when the model relative humidity drops below 30 %, which is indicative of when a trajectory is in the free-troposphere. So for example, the back trajectory that ends at 500 m within the POC had remained in the boundary layer for the previous four days and travelled from the south-east. This can be contrasted with the 1500 m trajectory (red), that had mixed down into the boundary layer earlier on the 5th September. It can be seen that this trajectory had originated over the biomass burning source region in continental Africa and crossed into the south-east Atlantic about 5 days earlier. Furthermore, this trajectory also tracked through the elevated plume of smoke observed by CALIPSO on the 3rd September (point B in Fig. 3 a and d). We also plot additional back trajectories that are initialised at an altitude just beneath the boundary layer inversion at 12 hourly positions back along the black 500 m trajectory, in order to examine the time-history of where the free-tropospheric air that is entrained into the boundary layer originates from. The orange (T-12), dark blue (T-24), purple (T-36) and light blue (T-48) stars are these additional trajectory start points. The start height of each of these is adjusted to account for the lowering boundary layer depth to the south. The trajectories that begin after midday on the 4th September (red, orange and dark blue) have all mixed down free-tropospheric air into the boundary layer that has been transported from the north and east, where the CALIPSO lidar data indicates the presence of an extensive biomass burning aerosol plume. Prior to this time, the trajectories (purple and light blue) switch to mixing in free-tropospheric air that has originated from the more pristine free-troposphere to the south-east. This dramatic change provides additional support to the idea that the boundary layer airmass in which the POC formed in the early hours of the 4th September would not have made contact with overlying biomass burning aerosol until it had moved further northwards towards Ascension Island.

## 4 Flight track

Figure 4 shows the horizontal and vertical flight patterns performed during flights C051 and C052 on the 5th September 2017. Flight C051 performed measurements in the overcast cloud field downwind of the POC in the morning (08:58:55 to 12:13:01 UTC). The cloud conditions were the same as those immediately to the east of Ascension Island that are shown in Fig 1d. The flight pattern included deep profiles to measure the boundary layer and free-tropospheric aerosol and thermodynamic structure



and straight and level runs at several altitudes, including within the cloud layer. In addition, several shallow profiles were performed to measure the vertical cloud structure beneath the trade-wind inversion. Flight C052 then transited to the south-east in order to perform measurements within the POC in the afternoon (14:09:02 to 17:38:13 UTC). The lower panel on Fig 4 includes the names of several profiles (P0 to P10) and a level run (R1) that will be referred to throughout the manuscript.

The flight pattern consisted of an initial deep profile to an altitude of 7150 m (profile P0), followed by a high-level run to about 8.8° W. On this run, cloud top heights were measured with the lidar and a series of dropsondes were released. This was followed by a profile descent into the POC (profile P1). A series of vertical profiles were then performed on the return leg back towards Ascension Island (profiles P2 to P10). These spanned altitudes from 35 m above the sea-surface to about 2250 m. This enabled aerosol, cloud and thermodynamic measurements to be made throughout the depth of the boundary layer and

across the trade-wind inversion into the lower free-troposphere. The series of vertical profiles on the return leg was interrupted for a level run (R1) at about 1320 m altitude in order to make additional cloud measurements. The cloud conditions on flight C052 were similar to that shown in Fig 1e. The overcast cloud downwind of the POC that was measured on the morning flight had cleared by the afternoon, making it difficult to visualize the northern edge of the POC feature in the satellite imagery. The in-situ measurements and back trajectories shown in Sect. 6, do however demonstrate that the return leg sampled the airmass

both within the POC and downwind of the open cell region.

  Whilst the majority of the aircraft observations in this study are from the two flights on the 5th September 2017, we do also briefly examine data from additional flights made on the 6th September 2017. Data from these latter flights in the vicinity of Ascension Island (flight numbers C053 and C054) are used for comparison with the LASIC surface measurements.

## 5 Aerosol and thermodynamic vertical structure

Figure 5 shows examples of the aerosol and thermodynamic vertical structure measured both downwind of and within the POC. These include aerosol number concentration from the PCASP and CPC, black carbon mass concentration from the SP2, carbon monoxide, water vapor mixing ratio and potential temperature. Data from the deep profiles immediately after take-off on flights C051 and C052 are included to illustrate the temporal change between the morning and afternoon at Ascension Island. These profiles can be contrasted to measurements made on flight C052 within the middle of the POC feature. Both of the downwind

profiles of moisture and temperature show a decoupled boundary layer, with a fairly well-mixed layer between the surface and the lifting condensation level (LCL), which is at an altitude of about 600 to 700 m. Above the LCL, there is another well-mixed layer that extends up to the trade-wind inversion at the top of the boundary layer, which is located at an altitude of about 1.8 km. There are strong and vertically shallow gradients (∼ 50 m) in both temperature and moisture at the top of the boundary layer. The thermodynamic profiles suggest that the overcast cloud measured on the morning of the 5th September on flight C051

downwind of the POC (Fig 1d) consisted of a stratocumulus layer that was decoupled from the surface. The LASIC vertically pointing radar and radiosondes do however suggest that coupling between the surface mixed layer and the stratiform cloud may have occurred intermittently via shallow cumulus (see Fig 14). In contrast, the POC moisture and temperature profile shows a well mixed sub-cloud layer from the surface to the LCL, followed by a conditionally unstable layer that extends to the base of





the trade-wind inversion. This is more typical of a shallow cumulus boundary layer profile as would be expected in the open cell region. The temperature beneath the LCL on the POC profile is notably cooler than the profiles downwind, which could be indicative of cooling via rain evaporation. The height and strength of the trade-wind inversion in the POC is very similar to the downwind profiles.

Figure 6 summarizes the cloud top height vertical profile measured using the aircraft lidar when flying above the boundary layer. Data from flight C051 are taken from the two deep aircraft profiles and a high level run. They are representative of the stratiform conditions downwind of the POC in the morning, at about 09:30 UTC. Data from flight C052 are taken from the outbound high level leg in the afternoon over the POC at about 15:00 UTC. The plot shows the frequency of cloud top height measurements, using only those lidar returns that detect cloud. This corresponds to 100% of the lidar returns on flight C051

and 88% of those on flight C052, with the lower frequency of cloud detection on flight C052 reflecting the more broken cloud conditions in the POC. This is broadly comparable to the SEVIRI cloud fractions shown in Fig. 2 a, which at the time of the aircraft measurements in the POC vary between $\sim 75$ and 90 %. We expect the lidar to be at the upper end of this range, given that it is likely to better detect optically thin stratiform clouds at the top of the boundary layer that may be detrained remnants of the more active cumulus, such as the example photos shown in Fig. 7 and the in-situ measurements presented in

Sect. 8. The lidar data are then binned into 100 m altitude bins. The measurements show a fairly invariant cloud top height over the stratiform cloud measured on flight C051, with a mean and standard deviation of $1836 \pm 50$ m. In contrast, the data over the POC on flight C052 shows a bi-modal distribution, with a peak in the cloud top height distribution in the 1.7 to 1.8 km altitude bin that corresponds to the approximate height of the trade-wind inversion. Cloud top heights in the POC do however continue to extend down towards the LCL at about 600 m, which results in the mean being markedly lower and a larger standard

deviation in cloud top height over the POC ($1599 \pm 287$ m.) than that measured over the stratocumulus cloud. This bi-modal distribution in cloud top height in the POC is typical of open cellular cloud conditions (Muhlbauer et al. , 2014).

The profiles of accumulation mode aerosol number concentration measured with the PCASP in Fig 5a show a large plume of aerosol in the free-troposphere that pervades across both the downwind and the POC cloud regimes. This is consistent with the satellite retrievals of above cloud aerosol optical depth shown in Fig 2. The in-situ measurements show that the aerosol

plume is in contact with the top of the boundary layer and extends upwards to about 4 km altitude. The concentrations of aerosol directly above the trade-wind inversion are in excess of 1000 cm$^{-3}$. In the boundary layer, there is a marked contrast in the aerosol conditions between the downwind profiles and the profile within the POC. Both profiles downwind of the POC show polluted conditions in the boundary layer, with mean PCASP concentrations beneath the trade-wind inversion and above the LCL of 460 and 225 cm$^{-3}$ on the morning and afternoon flights (C051 and C052). Between the surface and LCL, the

PCASP concentrations are lower at 275 and 110 cm$^{-3}$ for C051 and C052. The vertical profile of the aerosol in the marine boundary layer downwind of the POC is consistent with the decoupled boundary layer structure. The free-tropospheric aerosol is initially entrained downwards across the trade-wind inversion and then mixes down across the LCL into the surface mixed layer. In the POC, the PCASP concentrations in the surface mixed layer are much lower than in the downwind profiles, with a mean value of 28 cm$^{-3}$. This drops off to pristine values of 1 to 2 cm$^{-3}$ between 1.2 km and the altitude of the trade-wind

inversion. This ultra-clean layer (UCL) and the low sub-cloud aerosol concentrations are typical of previous measurements





made in POCs (Terai et al. , 2014) and in open cells at the stratocumulus to cumulus transition (Abel et al. , 2017; Wood et al. , 2018) and result from efficient removal of aerosol via the collision coalescence process (O et al. , 2018). It is remarkable that the PCASP concentration across the trade-wind inversion in the POC changes by over three orders of magnitude, suggesting that entrainment of overlying aerosol into the POC is not an efficient process.

The contrast between the more polluted boundary layer conditions downwind of the POC and the much cleaner boundary layer in the POC are also highlighted by the measured profiles of carbon monoxide, BC mass concentration and CPC number concentration shown in Fig 5. All of these measurements are significantly enhanced in the free-troposphere across both cloud regimes, with the elevated CO and BC consistent with that aerosol originating from biomass burning sources. Data from an Aerosol Mass Spectrometer (not shown) also confirms that this plume contains significant organic aerosol loadings. In addition,

these measurements all correlate with the PCASP concentrations in the boundary layer downwind of the POC, indicating that biomass burning aerosol has been mixed across the trade-wind inversion. This is contrasted with the carbon-monoxide data beneath the trade-wind inversion in the POC, which is invariant in altitude and typical of clean marine conditions measured at Ascension Island during the LASIC deployment (Pennypacker et al. , 2019). Carbon-monoxide is a relatively long-lived tracer of the biomass burning aerosol that is not significantly affected by precipitation, in contrast with the aerosol particulates

that can be removed efficiently via collision-coalescence processes. The BC mass concentrations in the POC are also low, providing additional evidence of limited mixing of smoke into the POC. Finally, the peaks in the concentration of aerosol particles measured with the CPC in the cloud free UCL (see photograph in Fig. 7c) and the sub-cloud layer in the POC are indicative of recent new particle formation. Episodic increases in Aitken mode particles have often been observed in POCs from both aircraft and shipborne measurements (Petters et al. , 2008; Wood et al. , 2008; Terai et al. , 2014). The measurements

from this case are consistent with cloud resolving model simulations that include a detailed aerosol and chemistry scheme of a different POC case (Kazil et al. , 2011). In those simulations, the authors find that the observed DMS flux from the ocean can support a nucleation source of aerosol in open cells that exceeds sea salt emissions in terms of the number of particles produced. It is important to note that the observed new particle formation in the UCL would not occur if significant mixing of the free-tropospheric aerosol into the boundary layer was occurring as the pre-cursor gases would preferentially condense

onto those larger particles instead of forming new aerosols. These measurements therefore all indicate that the entrainment of overlying biomass burning aerosol may be significantly more efficient in the overcast cloud layer downwind of the POC, suggesting that the cloud regime may play an important role on controlling when free-tropospheric aerosol can mix into the boundary layer. It also demonstrates that the model winds on which the trajectories in Fig. 3 follow mix free-tropospheric air into the boundary layer too readily in the location where the POC was observed. This is perhaps not surprising, given that

the meteorological data used to calculate the trajectories is based on a global NWP analysis, which will not be capable of simulating complex mesoscale features such as POCs.





## 6   Locating the POC boundary at the time of the aircraft measurements

Figure 1 e showed that on the afternoon of the 5th September, the overcast stratiform cloud downwind of the POC had cleared, making it difficult to identify the location of the boundary between the POC and downwind conditions using satellite imagery at the time of the aircraft measurements. Here, we examine the aerosol and CO data along the return leg on flight C052 (15:42 to 17:26 UTC), when the aircraft performed saw-tooth profiles that spanned the depth of the boundary layer, from deep within the POC back towards Ascension Island. These in-situ measurements are shown in Fig 8 and will be used to identify where the marked change between a clean marine and more polluted boundary layer occur, that are associated with the POC and downwind conditions.

Figure 8a plots the altitude of the aircraft as a function of longitude. The dashed line is the initial profile out of Ascension Island and the solid line the low-level return leg. Profile and run names are indicated above the figure. The red crosses indicate the altitude of the base of the trade-wind inversion from every profile that crossed between the boundary layer and the free-troposphere. The location of the inversion base was identified visually from the temperature and humidity measurements. The majority of the inversion crossings are in accordance with the cloud top height measured with the lidar on the outbound high-level leg over the POC (Fig 6). The exception is the lowering of the inversion height on profiles P9 and P10 as the aircraft traveled west towards Ascension Island. This is associated with the large-scale cloud clearance in the afternoon downwind of the POC and was also seen in the surface based measurements shown in Fig 14.

The filled black circles in Fig 8a indicate where cloud or precipitation was measured, using a water content threshold of 0.01 g m$^{-3}$ on the Nevzorov TWC sensor. This shows that on take-off (profile P0), a single layer of cloud was measured just below the trade-wind inversion. On the return leg later in the afternoon, cloud and/or precipitation was observed at all levels beneath the trade-wind inversion east of 13° W. The cloud microphysics measurements will be examined in more detail in Sect. 7. However, given that the LCL was approximately 600 to 700 m as can be inferred from the water vapor profile in Fig 5e, it is evident that the Nevzorov measurements indicate that precipitation was frequently observed below cloud base down to the lowest altitude of the measurements (100 ft above sea level). To the west of 13° W, the cloud had cleared on the return leg.

The blue, green and red shaded altitude bands in Fig 8a are used to composite the aircraft measurements in the lower panels of the figure. They represent height ranges where all of the profiles were in the free-troposphere (blue), in the height range of the ultra-clean layer in the upper part of the decoupled boundary layer within the POC (green) and in the surface mixed layer beneath the LCL (red). Figure 8b to d show how carbon monoxide, accumulation mode aerosol number concentration and BC mass concentration varied as a function of longitude in the three altitude bands. All of these measurements show enhanced levels in the free-troposphere when compared to the boundary layer data, indicating that the elevated free-tropospheric biomass burning aerosol plume pervaded across the ~750 km horizontal distance covered by the aircraft on flight C052. The free-tropospheric concentrations gradually increase eastwards from measurements downwind of the POC to over the POC itself. Below the trade-wind inversion, the CO data show low clean background values ~ 70 ppb throughout the depth of the boundary layer to the east of 12° W. These values are in accordance with the background levels of CO measured from the LASIC measurement site on days that exhibit very clean aerosol conditions between June 2016 and October 2017 (median





CO concentration of 69 ppb, with an inter-quartile range of 62 to 74 ppb (Pennypacker et al. , 2019)). Further west, there is a gradual increase in CO concentrations towards Ascension Island, indicating that the airmass containing biomass burning aerosol has mixed into the boundary layer. It can also be seen that in the upper part of the boundary layer to the west of 13° W, the CO concentration is higher than at lower levels, in accordance with the example downwind profiles shown in Fig 5.

The PCASP number concentration in the upper part of the boundary layer (green) shows the ultra-clean layer ($N_{PCASP}$ of a few cm$^{-3}$) in the POC between 12 and 10° W. This region also shows the lowest PCASP concentrations at lower levels ($N_{PCASP}$ ranging from 15 and 29 cm$^{-3}$ between profiles P5 and the ascent up to R1). These low aerosol concentrations are indicative of the removal of aerosol by active collision-coalescence processes. Interestingly, the eastern-most profiles (P1 to P3) show more elevated aerosol concentrations, with median $N_{PCASP}$ concentrations ranging from 75 to 90 cm$^{-3}$ and 40 to

72 cm$^{-3}$ in the surface and elevated boundary layer height ranges. This is in spite of the low CO values, suggesting that the enhanced aerosol loadings to the east measured on profiles P1 to P3 as compared to the lower values on profiles P4 to P7 are not due to significant mixing of elevated biomass burning aerosol into the POC and are more likely due to enhanced aerosol loadings from the Ocean surface. This is further backed up by the fairly low BC mass concentration values measured by the SP2 that are shown in Fig 8d. They show that in the surface mixed layer, the BC mass is typically a few tens ng m$^{-3}$ to the

east of 12° W and that the BC mass loadings only increase significantly when the aircraft traveled west of 13° W. In addition, in the upper part of the boundary layer, BC mass loadings were below the detectable limit of the SP2 to the east of 12° W.

The correlation between increasing CO, $N_{PCASP}$ and BC mass to the west of 13° W indicates that mixing of elevated biomass burning aerosol into the boundary layer had been more prevalent at the westward end of the aircraft measurements. To understand if this westward increase occurs across the POC boundary and to explore possible reasons for the slight enhance-

ment in the boundary layer aerosol loadings on the eastern-most profiles, we examine satellite imagery in combination with trajectories initialized at the time and position of each profile (P0 to P10) and run (R1).

Figure 9 a and b show SEVIRI true-color imagery at 10 and 16 UTC on the 5th September. These times are broadly representative of the cloud conditions observed during the morning and afternoon flights C051 and C052. The location of each of the profiles and the level run on the return low-level leg on flight C052 are shown with colored stars. From each of these

positions, we calculate backwards and forward trajectories, that begin at 500 m altitude and follow the boundary layer airmass. Using these trajectories, the relative position of the profiles and run at the time of the satellite images is then marked with an open circle on the figure. As already discussed, the overcast cloud downwind of the POC cleared in the afternoon of the 5th September, making it difficult to define the POC boundary on the flight track of C052 (Fig 9 b). However, in the morning satellite image (Fig 9 a), the main POC feature is well defined and a secondary feature can also be seen in the south-east of

the image. These two POC features were labeled 'A' and 'B' in Fig 1. The open circles show that the airmass in which profiles P0, P8, P9 and P10 were made are in the overcast cloud downwind of the POC in the morning. Similarly, run R1 was close to the western POC boundary, profiles P5, P6 and P7 were deep within POC 'A', profiles P4 and P3 were close to or on the eastern boundary of POC 'A' and profiles P2 and P1 were in a region of overcast cloud that was situated between the two POC features 'A' and 'B'. By the afternoon image and the time of the aircraft measurements on flight C052, this region of overcast

cloud that had separated POC 'A' and 'B' had turned into open cells i.e. the two individual POCs had merged. We note that the





operational MODIS cloud top effective radius product shown in Fig 9 c and d shows that this region had values approaching 25 $\mu$m in the morning and was therefore more likely to form drizzle than the overcast cloud downwind of the POC, that had much smaller values $\sim 10\mu$m. We also note that the lower effective radius values to the north-west of the POC are where biomass burning aerosol had mixed down into the boundary layer.

This analysis is consistent with the longitudinal variation we see in the aircraft measurements in Fig 8. The increase in boundary layer aerosol to the west of run R1 is consistent with there being more efficient entrainment of free-tropospheric biomass burning aerosol into the region containing overcast cloud conditions downwind of the POC in the morning, than within the POC itself. Similarly, the increase in marine aerosol on the eastern-most profiles matches the location of the cloud that separated the two POC features in the morning, but that then fully developed in to open cells in the afternoon. We hypothesize

that the time for enhanced removal of boundary layer aerosol by collision-coalescence and sedimentation processes was less in the area of developing open cells than deep within the POC i.e. the airmass at the eastern extent of the measurements had experienced heavy precipitation for less time.

## 7   Compositing aerosol, cloud and thermodynamic data downwind of and within the POC

Based on this analysis, we now composite the vertical profiles from the aircraft data from both flights into conditions represen-

tative of the airmass downwind of the POC and conditions within the POC itself. The morning flight, C051, is representative of the downwind conditions only. For the afternoon flight, C052, we define measurements made to the west of 13° W as being representative of the downwind conditions and to the east of 12° W as being representative of the POC. This is essentially separating the data on the return low-level leg either side of run R1, which was roughly at the transition between the two cloud regimes. Run R1 is examined further in Sect. 8. We composite the data into 200 m altitude bins and calculate the median and

interquartile range from a variety of parameters.

    Figure 10 shows the composite vertical profiles of aerosol number concentration measured with the PCASP and CPC, carbon monoxide, potential temperature, water vapour mixing ratio and relative humidity. These profiles show the same broad features that were presented in the individual profiles in Fig 5. There is a free-tropospheric biomass burning aerosol plume that is prevalent in both regimes, with evidence that the smoke has mixed down into the boundary layer downwind of the POC. The

POC measurements show a very clean boundary layer, including the presence of an UCL between about 700 m and 1.8 km. There is significant variability in the CPC data in the UCL, which indicates that regions of elevated Aitken mode aerosol concentrations are rather heterogeneous. This could be due to enhanced droplet scattering close to the intermittent clouds in the POC leading to localized areas of increased actinic flux that can promote new particle formation, although this cannot be concluded from these measurements. There is a sharp temperature inversion and moisture gradient at the top of the boundary

layer, with the base of the inversion in the POC and downwind profile from the morning flight C051 located at 1.8 km. The downwind profiles from flight C052 clearly show a lowering of the inversion base to about 1.6 km, that is associated with the afternoon cloud clearance. The thermodynamic profiles show a decoupled boundary layer in both downwind profiles. However, the RH in the upper part of the boundary layer remains sub-saturated at all levels in the C052 downwind profile, suppressing





cloud formation. This can be contrasted to the conditions in the morning C051 downwind profile, where the deeper boundary layer is saturated in the uppermost 200 m. In the POC the thermodynamic profile is more typical of a shallow cumulus boundary layer. The relative humidity shows saturated conditions at all levels above the LCL, highlighting that the vertical extent of the clouds in the POC span a much larger depth than in the downwind profiles.

Figure 11 a to c shows composite vertical profiles of a selection of in-cloud parameters for the periods where cloud was observed, namely the measurements made within the POC on flight C052 and the downwind stratiform cloud measured on flight C051. The data in each altitude bin are calculated from in-cloud points only using a threshold liquid water content of 0.01 g m$^{-3}$ from the CDP to define cloud. There are striking microphysical differences between the two cloud regimes. The cloud drop number concentration, $N_L$, in the stratiform region downwind of the POC had values of 150 to 200 cm$^{-3}$.

Significantly lower values of $N_L$ were measured in the POC itself. These peaked at $\sim 25$ cm$^{-3}$ at 750 m altitude and then dropped off to less than 10 cm$^{-3}$ higher up in the boundary layer. This contrast between the two cloud regimes broadly mirrors the change in accumulation mode aerosol concentration measured with the PCASP shown in Fig 10 a, that can be used as a rough proxy for the CCN concentration. The change can be attributed to both mixing of biomass burning aerosol into the boundary layer downwind of the POC acting to increase $N_L$ and active collision-coalescence processes in the POC acting

to reduce $N_L$. The volume mean radius of cloud drops calculated from the CDP size distribution, $r_{V,L}$ shows similar large differences, with values of 5 $\mu$m typical in the downwind stratifom cloud and 10 to 20 $\mu$m in the POC. It is notable that the size of the cloud drops tends to increase with altitude in the POC. The in-cloud liquid water content profiles, $q_L$, show an adiabatic looking increase in cloud water in the stratiform region and more variability in the POC. However, the in-cloud $q_L$ in the POC is higher than in the downwind stratiform cloud field as might be expected from the increased depth of the clouds in the POC.

Figure 11 d shows how the cloud fraction varies with altitude from the composite profiles. The cloud fraction is calculated from the ratio of the number of in-cloud data points divided by the total number of data points in each altitude bin. For the downwind stratiform cloud region, the cloud fraction peaks at 0.8 at the top of the boundary layer. The profile in the POC shows lower values as expected given the more cumuliform nature of the clouds, with values of about 0.2 between 500 m and 1 km altitudes. The cloud fraction does however tend to increase with height up to 0.6 at an altitude of 1.5 km in the POC.

This may reflect the increased amount of detrained cloud layers that surround the individual cumulus that are present in the UCL, examples of which can be seen in the photographs shown in Fig 7. Also included in Fig. 11 d are the profiles of rain fraction. Rain points are defined as those points where the concentration of drops larger than 60 $\mu$m as calculated from the 1Hz composite PSD exceeds 1 L$^{-1}$. The rain fraction profiles show that large drizzle or precipitation particles were not observed below cloud base in the downwind stratiform cloud region. In the POC, precipitation sized particles were prevalent throughout

the depth of the cloud layer and also were observed down to the surface.

    Composite profiles of various precipitation averaged parameters are shown in Fig 12 using the same 1 L$^{-1}$ threshold on large drops to define in-rain data points. Each of the parameters are calculated from the composite PSD using data for particles larger than 60 $\mu$m. They include the rain drop number concentration, $N_R$, the raindrop volume mean radius, $r_{V,R}$, the rain water content, $q_R$, and the rain rate. The fallspeed relation of Beard (1976) is used in the calculation of rain rate. It is immediately

apparent that the values of $N_R$ in the upper part of the boundary layer are an order of magnitude higher in the POC than in





the downwind stratiform cloud region and that only in the POC does the precipitation fall to the surface. The $r_{V,D}$ in the POC gradually increases and the $N_R$ decreases from the top of the boundary layer down to the LCL, even though the rain fraction remains fairly constant (Fig 11 d). This is consistent with the largest rain drops falling to lower levels in the cumulus clouds as their fallspeed is more likely to exceed the cloud updraft speed than that of smaller drizzle sized drops. As they fall through

the cloud they will also continue to grow via the accretion of cloud drops. The largest rain drops and precipitation rate are found below cloud base, with median peak values reaching several tens of mm day $^{-1}$. The profiles clearly demonstrate that precipitation is prevalent within the POC. Although low concentrations of drizzle sized drops were observed in the stratiform cloud downwind of the POC, the rain-rate calculated from integrating the size distributions is negligible ($\sim 0.01$ mm day$^{-1}$) and confined to the cloud layer.

## 8   UCL clouds and the transition region

There has been a growing interest in clouds that form in the low aerosol environment found in UCLs, especially with regards to the quasi-laminar stratiform cloud layers that are likely to be detrained remnants of more active shallow cumulus (Wood et al. , 2018; O et al. , 2018). These more stratiform layers, also termed veil clouds, have been characterised in the stratocumulus to cumulus transition in the north-east Pacifc (Wood et al. , 2018) and are frequently observed in POCs (Wood et al. , 2011;

Terai et al. , 2014). Wood et al. (2018) show that these clouds tend to exhibit low levels of turbulence, are often both vertically and optically thin and exhibit low concentrations of large liquid drops. They were also a common occurrence in the POC case study presented here as illustrated by the example photographs in Fig. 7. The photographs show vertically thin stratiform layers that had formed in the upper part of the decoupled boundary layer and are distinct from the cumulus clouds. Whilst it is not possible to determine what contribution these thin stratiform clouds have on the composite cloud and precipitation vertical

profiles shown in Figs 11 and 12, data from the straight and level run R1 that was located close to the POC boundary nicely illustrates the contrasting microphysical conditions between a quiescent cloud and a more active cumulus cloud in the UCL.

Figure 13 shows selected aircraft data on the level run R1, which was flown at an altitude of 1320 m. This altitude is in the middle of the UCL and the run was located close to the boundary between the POC and downwind airmass, with the data on the right of the plot being farther east and therefore deeper into the POC (see also Fig. 8). Figure 13 a plots the air vertical velocity

along the run. The two grey shaded bands are selected to illustrate contrasting cloud dynamical environments. At about 12.5° W the aircraft flew through some active cumulus clouds with peaks updrafts approaching 4 m s$^{-1}$, downdrafts of 1 m s$^{-1}$ and a vertical velocity variance calculated from 32 Hz wind measurements $\sigma_w^2$ of 1.10 m$^2$ s$^{-2}$. The other segment of the run centred at 12.15° W shows a contrasting cloud environment that is much more quiescent ($\sigma_w^2$ of 0.03 m$^2$ s$^{-2}$), in accordance with the low turbulent conditions that are common in UCL veil clouds (Wood et al. , 2018). The out of cloud PCASP values plotted in

panel c demonstrate that to the east, the environment that the quiescent cloud formed was in the UCL with low accumulation mode aerosol concentrations $\sim 1$ to 10 cm$^{-3}$. On the western edge of the more active cumulus, this increases to 40 to 50 cm$^{-3}$. This could be indicative of mixing of the clean UCL air with the more polluted boundary layer air across the POC edge due to local circulations induced from precipitation and dynamical feedbacks at the open-cell boundary (Wood et al. , 2008; Wang





and Feingold , 2009b). Another hypothesis could be that the active cumulus can penetrate across the trade-wind inversion and locally mix down some free-tropospheric biomass burning aerosol into the boundary layer. Even if this mechanism did occur, the prevalence of low CO values in the UCL in the POC shown in Fig. 10 suggests that it does not dominate the aerosol budget within the POC.

Figure 13 b shows the liquid water content along the run calculated from the CDP which measures drops $< 50\ \mu$m. Also included is data from the Nevzorov TWC sensor and from integrating the composite particle size distribution, both of which measure the cloud drops and precipitation sized particles. In the active cumulus clouds, the three measurements are in broad agreement, indicating that a large fraction of the LWC is contained in the cloud drops. This can be contrasted to the more quiescent cloud measured between 12.1 and 12.2° W. Here, the precipitation sized particles dominate the condensate as the

CDP measurements are significantly lower than the Nevzorov and composite PSD data. This contrast is also highlighted in the mean size distributions for these regions shown at the bottom of the figure. In the cumulus cloud, there is a pronounced cloud droplet mode that peaks between 14 and 40 $\mu$m diameter and a shoulder to higher sizes that contains the precipitation sized particles. In the more quiescent cloud, the cloud drop mode in the size distribution is not evident. We also overlay the mean in-cloud PSD measured in the stratocumulus clouds that were sampled within the more polluted boundary layer downwind of the

POC on flight C051 on Fig. 13 f and g. As expected from the composite analysis shown in Fig. 11, this shows a propensity for much higher concentrations of small cloud drops and an almost complete absence of large drizzle or rain drops when compared to the cumulus and quiescent cloud PSDs measured on run R1.

    The cloud drop number concentration along run R1 is plotted in Fig. 13 c and shows that the quiescent cloud region has very low values of a few cm$^{-3}$ that are in accordance with the low aerosol concentrations in the UCL, whereas the more

active cumulus clouds sampled have values of $\sim 40$ cm$^{-3}$. It is also evident that the concentration of precipitation sized drops increases from about 0.1 cm$^{-3}$ in the cumulus to almost 1 cm$^{-3}$ in the quiescent cloud region. These cloud microphysical contrasts are very similar to the data shown in veil clouds by Wood et al. (2018). O et al. (2018) perform idealised parcel model simulations to demonstrate that cloud drops in the active cumulus are efficiently removed via collision-coalescence processes, such that any detrained moist air would likely be devoid of cloud drops and also exhibit the very low CCN concentrations typical

of the UCL. The effective radius calculated using only cloud drops measured by the CDP and using the cloud plus precipitation sized particles in the composite PSD is shown in panel d. Rainfall rates calculated from integrating the composites PSD are included in panel e. It is clear that both cloud regions exhibit large rain rates in excess of 10 mm d$^{-1}$ and that the $r_{eff}$ can exceed 50 $\mu$m when the precipitation size drops are included. The significant rain rate in the quiescent cloud region suggests that this cloud would decay rather rapidly without further replenishment of liquid water. For example, taking a LWC of 0.25 g

m$^{-3}$ from the observations in the quiescent cloud region, the majority of which is contained within drizzle and rain sized drops and an assumed cloud thickness of 200 m, which is typical of the veil clouds studied in Wood et al. (2018), then a precipitation rate of 10 mm d$^{-1}$ would sediment out the condensate in $< 10$ min. As discussed in Wood et al. (2018), mechanisms that could contribute additional condensate into the quiescent cloud such as outflow from new active cumulus cells or from mesoscale ascent would likely be required to explain the several hour longevity that is often observed in UCL veil clouds.



## 9   The view from the LASIC ARM site

In this section, we shift the focus to looking at the downwind and POC conditions from the surface based ARM site on Ascension Island. The upper panel in Fig. 14 plots a time-series of radar reflectivity measurements from the vertically pointing Ka band radar, with cloud base height measurements from the ceilometer overlaid. The data span the 5th and 6th September.

On the morning of the 5th September the radar shows a thin layer of cloud sitting at the top of the boundary layer, with evidence of shallow cumulus clouds at lower levels that have a base at the LCL. Occasionally, enhanced radar reflectivities couple the lower cloud base with the upper stratiform cloud layer. This could be a sign of the cumulus rising into the more stratiform cloud above, or drizzle falling from the upper layer. The two cloud layer structure is supported by the double peak in relative humidity that is shown in both radiosonde and aircraft vertical profiles presented in the third panel of Fig. 14. It is

also consistent with the cloud top height derived from the cloud radar at the time of the soundings, that is also overlaid on that panel. A MODIS true color image at 10:44 UTC indicates that the cloud in the morning of the 5th September was stratiform in appearance and this time corresponds to the downwind measurements made by the aircraft on flight C051. On the afternoon of the 5th September, there is a large scale cloud clearance that can be seen in the radar and satellite imagery. The radiosonde relative humidity sounding also shows a lowering of the boundary layer depth and drying of the upper boundary layer that was

observed on the aircraft measurements made downwind of the POC on flight C052. At about 23 UTC on the 5th September the radar measures a rapid change from predominantly cloud-free conditions to clouds that exhibit significant radar returns. Some of these structures are vertically coherent and extend down to the minimum height detectable by the radar. It is likely that this is precipitation that is falling down to or close to the surface. This rapid change in cloud conditions late on the 5th September is consistent with the POC feature advecting over Ascension Island as determined from the trajectory analysis that was presented

in Fig. 1. The green trajectory in that figure was in the POC and was initialised at 00 UTC on the 6th September at Ascension Island. The radar data show that similar conditions were then measured at Ascension Island until late on the 6th September and the radiosonde and aircraft soundings show that the boundary layer structure was more typical of a shallow cumulus boundary layer. The associated cloud conditions can also be seen on the MODIS imagery from the 6th September in Fig. 14. The images suggest that the southern boundary of the remnants of the POC feature was roughly aligned west-east and located just to the

south of Ascension Island.

Finally, the middle panel of Fig. 14 shows a time-series of BC mass concentration, carbon monoxide and CCN concentration measured at the surface. Overlaid are measurements made from take-off or landing at Ascension Island from the aircraft averaged over the lowest 500 m of the boundary layer. This is generally in the surface mixed layer and so is broadly comparable to the airmass at the ARM site. Note that the aircraft PCASP concentration is included rather than CCN concentration. The

ARM data are consistent with the aircraft measurements presented previously, showing that downwind of the POC (before 23 UTC on the 5th September) there were elevated levels of aerosol, BC and CO at the surface, indicating that biomass burning aerosol had been mixed from the free troposphere down into the boundary layer. As the POC reached Ascension Island, there is a rapid reduction in all of these measurements. In particular, the low carbon monoxide measurements in the POC that serve as a good tracer for the continental airmass that transports the smoke over the Ocean, again suggests that the POC must be





less efficient at mixing the overlaying aerosol across the boundary layer inversion. We note that later in the day (18 UTC 6th September), there are small increases in the LASIC measurements of CO, BC and CCN concentration that indicate the presence of biomass burning aerosol at the surface whilst the cloud is still actively producing precipitation. We hypothesize that this is showing evidence of the flow of boundary layer aerosol northwards from the more polluted stratiform region immediately to

the south of Ascension Island into the southern edge of the open cells, and that this exceeds the removal rate of aerosol from collision-coalescence and sedimentation processes. Aircraft data from a low-level flight leg to the west of Ascension Island on the morning of the 6th September confirm that there is a north-south horizontal gradient in boundary layer aerosol and CO across the transition in cloud regime, with the cleanest airmass deeper within the open cell region to the north (see Fig. 15).

## 10   Open cell frequency of occurrence

The airborne and ground-based observations have shown a striking contrast between the polluted boundary layer in the closed-cell region downwind of the POC and the much cleaner conditions in the open-cell conditions within the POC, despite the fact that an extensive free-tropospheric biomass burning aerosol plume was in contact with the trade-wind inversion in both cloud regimes. Importantly, the low background CO values in the POC suggest that the open cells are not efficient at entraining large amounts of free-tropospheric aerosol into the boundary layer. If this was typical of boundary layers containing organized open

cells in general, then it is of interest to consider how frequent these conditions occur in the south-east Atlantic. Muhlbauer et al. (2014) use an artificial neural network cloud classification scheme to identify open and closed cells from one year of MODIS Aqua data. They find that over an area containing the semi-permanent stratocumulus cloud deck in the south-east Atlantic, the monthly mean frequency of occurrence of open and closed cell conditions ranges from about 20 to 12 % and 32 to 50 % respectively, for August through to October 2008 (estimated from their figure 6). This suggests that open cell conditions form

a non-negligible fraction of the boundary layer cloud morphology during the months when both large boundary layer cloud fractions and the episodic transport of biomass burning aerosol over the south-east Atlantic are most prevalent (Adebiyi et al. , 2015).

Here, we extend the work of Muhlbauer et al. (2014) to cover a 19 year period between 2000 and 2018, by examining the areal coverage of organized open cells in the south-east Atlantic during September, when the horizontal extent of elevated

aerosol optical depths from biomass burning are most extensive (Adebiyi et al. , 2015) and the cloud LWP reaches a maxima (Zuidema et al. , 2016b). To do so, we manually locate mesoscale open cellular regions from MODIS Terra day-time (10:30 equator crossing time) true-color imagery obtained from NASA Worldview. The images cover a longitude range of 15° W-14° E and a latitude range of 1-25° S. For each of the 570 individual daily images in this period, areas that exhibit open cell characteristics (dark cloud-free cellular regions surrounded by bright narrow cloud edges) are identified. Figure 16 a and

b show two examples, with the hand-drawn red lines outlining regions of open cells. The 18th September 2015 case shows an extensive area of open cells to the west of 5° E, that covers a large fraction of the south-east Atlantic. In contrast, we identify four distinct but smaller regions of open cells in the 5th September 2017 case. The northern two regions in the image correspond to the POC features studied in this paper. Further south, there are two more linear features that are roughly aligned



west-east. These types of features typically originate in the mid-latitudes and often move northwards through the stratocumulus cloud deck and are then more similar to the 'rift' features shown in Sharon et al. (2006) and Wood et al. (2008). A larger set of illustrative examples for the whole of September 2010 are shown in Fig. 17, which includes both a variety of POCs and more extensive regions of open cells. The open cell fraction for each of the daily images is then determined as follows. The

manually identified open cell regions are filled with a known colour (red) that is not present in the original image. We can then calculate the open cell fraction from the image as $P_{RED}/(P_{ALL} - P_{BLACK})$, where $P_{RED}$ is the number of red pixels in the modified image, $P_{ALL}$ is the total number of pixels in the image and $P_{BLACK}$ is the number of black pixels in the image. The black pixels in the image correspond to missing data e.g. due to the gaps that occur between adjacent satellite swaths in Fig. 16 a and b. The open cell fraction can also be calculated for any sub-region of the image that is given by a set of latitude and

longitude points. Whilst our analysis of open cell conditions from the satellite imagery is subjective, it is instructive to make a basic comparison to the results of Muhlbauer et al. (2014) in order to check for consistency. When looking at similar regions for September 2008, the method used in this study calculates a mean open cell fraction of 0.13 (10° W-10° E, 10°-25° S), which can be broadly compared to the Muhlbauer et al. (2014) value of 0.15 (estimated from their figure 6) (10° W-10° E, 10°-30° S). Given that there is a false detection rate of approximately 10–15 % in the neural network algorithm employed by

Muhlbauer et al. (2014) and that we expect that our method could miss some smaller regions of open cells, we consider this agreement to be satisfactory.

Figure 16 c shows a map of the mean September 2000-2018 open cell fraction, calculated from all of the individual daily images. It is evident that open cells are not common in the near coastal region and that their prevalence increases offshore, co-inciding with less-stable and deeper marine boundary layers. This is similar to an analysis of open cell locations in the

south-east Pacific (Wood and Hartmann , 2006). Peak open cell fractions in excess of 0.25 are found in an area to the south of Ascension Island (centred at 13° S, 10° W). Also shown in Fig. 16 c with grey shading is the area where the fine mode aerosol optical depth exceeds 0.2, which we take as a proxy for the location of where extensive biomass burning aerosols are often present at this time of year. The 0.6 total cloud fraction contour for liquid-phase clouds is also included (dashed black line) and corresponds to the region of the south-east Atlantic that typically exhibits extensive stratocumulus cloud cover. Both the

fine mode aerosol optical depth and liquid cloud fraction are mean values from September 2000-2018 and are taken from the MODIS Terra level 3 product. Although the aerosol optical depth product gives no information on where the biomass burning aerosol is located in the vertical, it is evident that the maxima in open cell fraction does co-incide with areas of high boundary layer cloud coverage and elevated biomass burning aerosol loadings. It is therefore plausible that subsiding free-tropospheric biomass burning aerosol layers transported from the continent may often come into contact with regions exhibiting open

cellular cloud morphologies.

We approximate the area of high boundary layer cloud coverage and elevated biomass burning aerosol loadings with the blue shaded area in Fig. 16c. Figure 16 d then shows the daily open cell fraction and the corresponding open cell area calculated in this region of the south-east Atlantic for September 2000-2018. The box and whiskers summarize the individual monthly data (the median, inter-quartile range and extremes). The mean September open cell fraction is 0.10, which corresponds to an area

covered by open cells of approximately 400,000 km$^2$. However, there is significant variability between different years and on





sub-monthly time-scales. For example, September 2002 had extensive periods devoid of open cells, whereas organized open cell convective structures were apparent every day in September 2010. Open cell fractions in excess of 0.1, 0.2, 0.3 and 0.4 occurred 41, 16, 4 and 1 % respectively during the 19 year period.

## 11    Conclusions and discussion

This work describes a case-study of a POC in the south-east Atlantic that was measured during the CLARIFY and LASIC field experiments. A combination of satellite data, model trajectories and in-situ measurements are used to describe the evolution of the POC feature. These suggest that the POC likely formed in a clean marine airmass, with an elevated free tropospheric biomass burning aerosol plume above that was not in contact with the boundary layer. After formation, the MBL winds advected the POC into a region where the base of the biomass burning aerosol had lowered. The in-situ observations show

that this aerosol was in contact with the trade-wind inversion in both the surrounding overcast cloud field downwind of the POC and within the POC itself.

The aircraft and surface based observations then demonstrate that the airmass downwind of the POC had entrained this overlying biomass burning aerosol, with enhanced accumulation mode aerosol concentrations, black carbon mass loadings and carbon monoxide present in the boundary layer. There was a marked contrast across the transition into the POC itself. Data

within the POC showed that the boundary layer was very clean, with low carbon monoxide and BC mass loadings and the presence of an ultra-clean layer immediately beneath the trade-wind inversion, in which evidence of new particle formation was observed. It is striking that within the middle of the POC, the accumulation mode aerosol concentration increased from a few $cm^{-3}$ directly beneath the trade-wind inversion to in excess of $1000\,cm^{-3}$ directly above. The cloud observations presented show that the open cells and more quiescent layer clouds within the POC are very clean and exhibit significant precipitation,

whereas the more polluted closed cells downwind have higher cloud drop concentrations and consequently very few drizzle sized drops. As carbon monoxide is a fairly long lived tracer of the biomass burning aerosol airmass that is not readily removed via cloud processing or precipitation, the low values in the POC that are typical of background values measured at Ascension Island at this time of year (Pennypacker et al. , 2019) indicate that it is not simply the enhanced rainfall in the POC that results in a cleaner boundary layer. All of these features therefore suggest that the organized open cellular convection in the POC is

very inefficient at entraining the overlying smoke into the marine boundary layer. The reduced efficiency in the mixing of free-tropospheric aerosols down into open cell boundary layers is consistent with previous inferences made from measurements of POCs in the south-east Atlantic (Wood et al. , 2011; Terai et al. , 2014) and from observations of the stratocumulus to cumulus transition in cold-air outbreaks (Abel et al. , 2017). We note however that these former studies have exhibited significantly lower free-tropospheric accumulation mode aerosol concentrations in contact with the inversion above the UCL ($\sim 10$ to $100$

$cm^{-3}$) than were observed from the measurements on this case-study. The low entrainment across the boundary layer inversion in the POC is also consistent with arguments made by Bretherton et al. (2010) and latter cloud-resolving model studies that demonstrate a much weaker entrainment rate within POCs compared to the surrounding overcast cloud field (Berner et al. , 2011, 2013).





This possible cloud regime dependence in entrainment has some important consequences for aerosol-cloud interactions in the south-east Atlantic. For example, Large Eddy Simulation (LES) studies of open cells that increase the MBL CCN concentration show microphysical and dynamical responses that can ultimately manifest in an increase in the scene albedo (Wang and Feingold , 2009b). The apparent low susceptibility of the open cells in this case to free-tropospheric aerosol intrusions prohibits

the biomass burning aerosol acting as a source of additional CCN. More broadly, the extensive reservoir of free-tropospheric biomass burning aerosols that are transported over the Ocean during the biomass burning season (Adebiyi et al. , 2015) are frequently observed to be in contact with the marine boundary layer (Lu et al. , 2018). Where they mix into the boundary layer will ultimately control when they can modulate the cloud evolution via microphysical perturbations. Our classification of open cellular convection in the region from 19 years of September MODIS imagery shows that the maxima in the open cell

frequency of occurrence ($> 0.25$) occurs far offshore. Although there is significant daily and inter-annual variability, the mean areal coverage of open cells is $\sim 400,000$ km$^2$ in an area of high mean liquid cloud fraction ($> 0.6$) and fine mode aerosol optical depth ($> 0.2$). It is therefore plausible that subsiding free-tropospheric biomass burning aerosol layers transported from the continent may often come into contact with regions exhibiting open cellular cloud morphologies. This aerosol-cloud "contact" has often been used as a proxy for investigating the cloud response to mixing of smoke into the boundary layer

from spaceborne measurements, with the assumption that these overlying aerosols are modulating the cloud microphysics (Costantino and Breon , 2013; Painemal et al. , 2014). Yet even in closed cell conditions, the timescale for mixing of elevated smoke into the MBL can be $\sim$ days and so instantaneous observations of aerosol-cloud "contact" are not the complete picture (Diamond et al. , 2018). The results presented here further demonstrate that additional care needs to be taken when interpreting indirect aerosol effects from observations of above cloud aerosols in open cell regions.

In addition, global weather and climate models are generally not capable of adequately simulating mesoscale open cell features such as POCs, due to both their coarse horizontal grid-spacing and often relatively simplistic representation of aerosol-cloud-precipitation interactions e.g. Abel et al. (2010). For example, it is clear that the trajectory model used for this study continued to readily mix free-tropospheric air down into the boundary layer airmass that contained the POC, in spite of it using meteorological data from a model analysis that had a much finer grid spacing than is typically employed in current climate

models. These larger scale models may therefore have significant errors in both the timing and the location of biomass burning aerosol induced cloud microphysical perturbations, with direct implications for estimates of the aerosol-indirect effect in the south-east Atlantic. The ability of models to simulate open cells is also important for determining the sign and magnitude of the direct effect of biomass burning aerosols, which is highly sensitive to the underlying reflectance (Abel et al. , 2005; Chand et al. , 2009) and therefore largely determined by the aerosol vertical distribution and cloud fraction. Evaluation of the spatial

and temporal frequency of open cells and an assessment of when smoke is mixed into the MBL in global climate models would therefore be worthwhile for future study.

It is however important to bear in mind that the measurements presented in this study are from a single case and additional observations of open cell conditions with free-tropospheric aerosol plumes in contact with the boundary layer are needed to confirm if these findings are typical. It would also be worthwhile to employ Large Eddy Simulation models to provide a more

mechanistic view of how the entrainment of overlying aerosols into POCs and the surrounding overcast cloud field differ.

*Data availability.* The FAAM aircraft data are archived at the Centre for Environmental Data Analysis (CEDA). The LASIC data were obtained from the Atmospheric Radiation Measurement (ARM) User Facility, a U.S. Department of Energy (DOE) Office of Science user facility managed by the Office of Biological and Environmental Research. ARM. The Terra MODIS L3 dataset was acquired from the LAADS Distributed Active Archive Center. The CATS and CALIPSO data were acquired from the NASA Langley Research Center Atmospheric Science Data Center (ASDC). AERONET AOD data are available from the AERONET web site (https://aeronet.gsfc.nasa.gov, last access: 14 June 2019). The Met Office St Helena radiosonde data and the NAME trajectory data are available from the lead author on request. Data from the SEVIRI cloud property and above cloud AOD retrievals are available from Fanny Peers on request.

*Author contributions.* SJA conceived the study and analysed the data with help from PB, PZ, JZ and MC. FP generated the SEVIRI above cloud AOD and cloud property dataset. JT provided the calibrated FAAM SP2 data. MF operated the FAAM SP2 instrument. IC and KB operated the FAAM 2DS and provided the data. SJA wrote the paper and all authors provided revisions and commentary on the manuscript.

*Competing interests.* The authors declare that they have no conflict of interest.

*Acknowledgements.* The CLARIFY deployment was jointly funded by the UK Natural Environment Research Council (NERC) through grant no. NE/L013479/1 and the Met Office. The LASIC field campaign was funded by the U.S. Department of Energy's Office of Science, Office of Biological and Environmental Research as part of the Atmospheric Science Research Program. We thank the whole CLARIFY and LASIC operations and science teams for their efforts on deploying and maintaining the instruments and for processing and calibrating the campaign datasets. We acknowledge the use of imagery from the NASA Worldview application (https://worldview.earthdata.nasa.gov/), part of the NASA Earth Observing System Data and Information System (EOSDIS). We thank Brent Holben for his effort in establishing and maintaining the Ascension Island AERONET site. FP was part-funded by the Research Council of Norway via the projects AC/BC (grant no. 240372) and NetBC (grant no. 244141). PZ and JZ acknowledge funding support from DOE ASR grant DE–SC0018272.



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

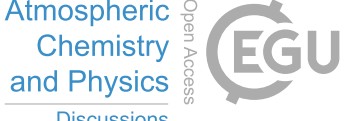



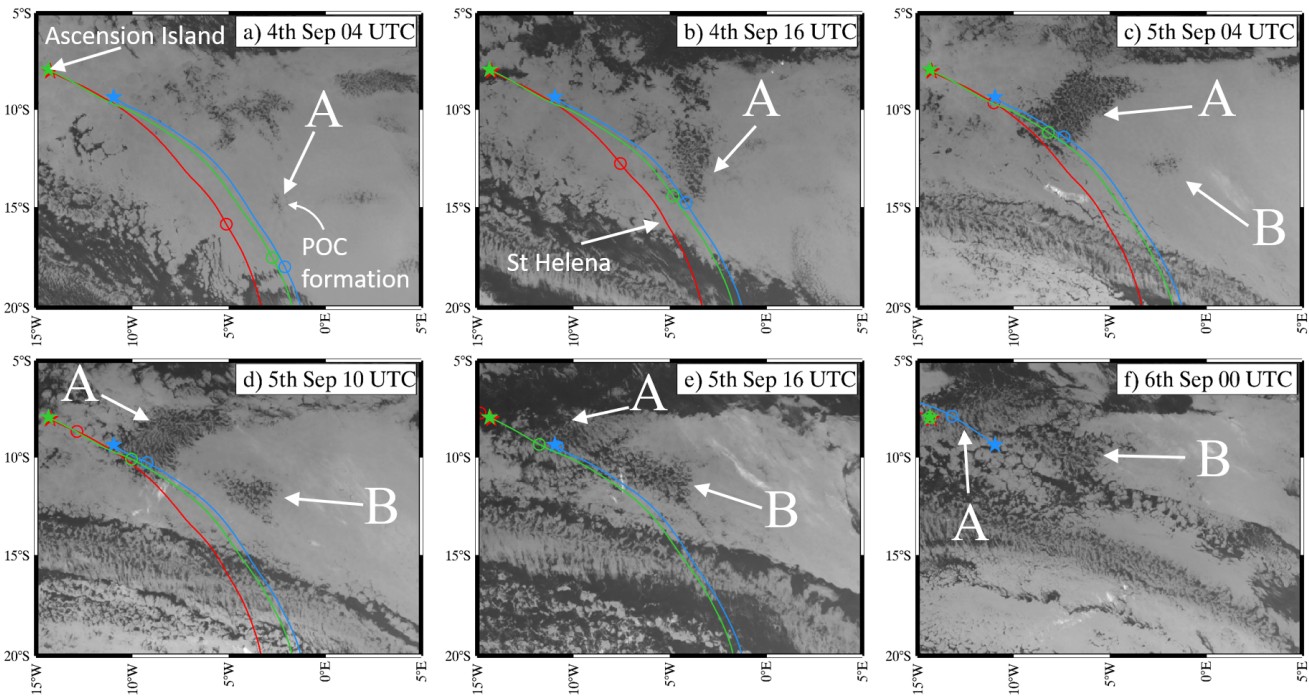

**Figure 1.** 10.8 $\mu$m brightness temperature imagery from SEVIRI showing the evolution of the POC feature as a function of time from the 4th to 6th September 2017. The main POC (labeled A) and a secondary POC feature (labeled B) are indicated. Three trajectories are overlaid on the imagery. These are initialised at 500 m altitude from the position and time of measurements made i) from the aircraft in the stratiform cloud region downwind of the POC (P0, red), ii) from the aircraft within the POC (P6, blue) and iii) from the LASIC site on Ascension Island within the POC (green). The stars on the trajectories show the position where each measurement was made and the open circles show the position along the trajectory at the time of each satellite image.



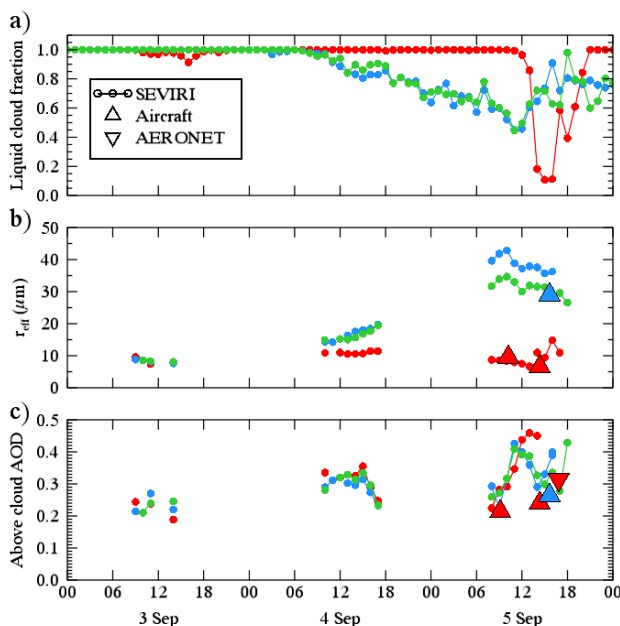

**Figure 2.** SEVIRI satellite data extracted from a 1×1 degree latitude-longitude box around the three trajectories shown in Fig 1. These are initialised from the position and time of measurements made i) from the aircraft in the stratiform cloud region downwind of the POC (red), ii) from the aircraft within the POC (blue) and iii) from the LASIC site on Ascension Island within the POC (green). Panel a) shows liquid cloud fraction, b) shows cloud top effective radius and c) shows above cloud aerosol optical depth. Aircraft data are overlaid in b) and c) from measurements made within (upward pointing blue triangles) and downwind (upward pointing red triangles) of the POC. Total column aerosol optical depth measurements made downwind of the POC from the AERONET site on Ascension Island are shown with a red downward pointing triangle in panel c).

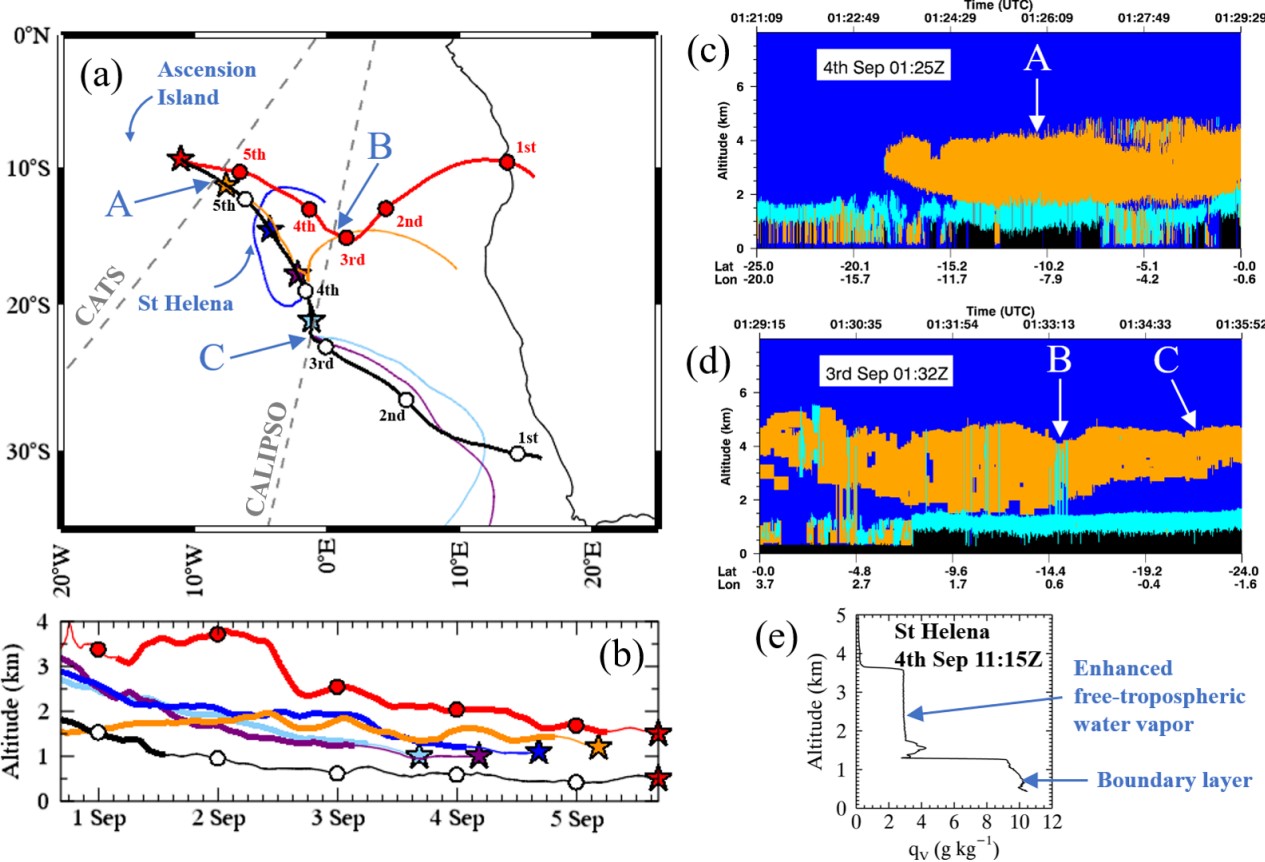

**Figure 3.** Examination of the airmass history of the POC. The black and red back tracks in panel a) are back trajectories initialised from the time and position of the aircraft observations in the POC (red star). The black trajectory was started at an altitude within the surface mixed layer (0.5 km) and the red trajectory beneath the boundary layer inversion (1.5 km). Additional back trajectories initialised at an altitude beneath the inversion at 12 hourly positions along the black 0.5 km trajectory are also shown. The orange, dark blue, purple and light blue stars are the trajectory start points. Panel b) shows the altitude above mean sea level of these trajectories as a function of time. The thick lines along each trajectory are where the model relative humidity < 30%, which is indicative of when the airmass was in the free-troposphere. Panels c) and d) show the lidar feature mask from CATS and CALIPSO respectively. Aerosol and clouds are shown with orange and cyan colors respectively. The satellite tracks are overlaid with a dashed line in panel a) and occasions where the lidar data are in the vicinity of the trajectories are labeled A to C. Panel e) shows a radiosonde profile of specific humidity from St Helena, launched at 11:15 UTC on the 4th September 2017. The date labels in a) and b) are valid at 00 UTC on each day.





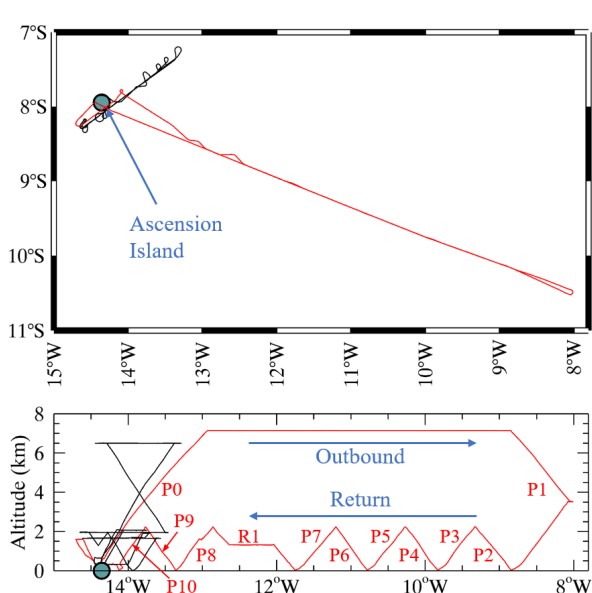

**Figure 4.** Flight track and altitude for flights C051 (black) and C052 (red). The location of selected vertical profiles (P0 to P10) and run (R1) on flight C052 are indicated on the lower panel.

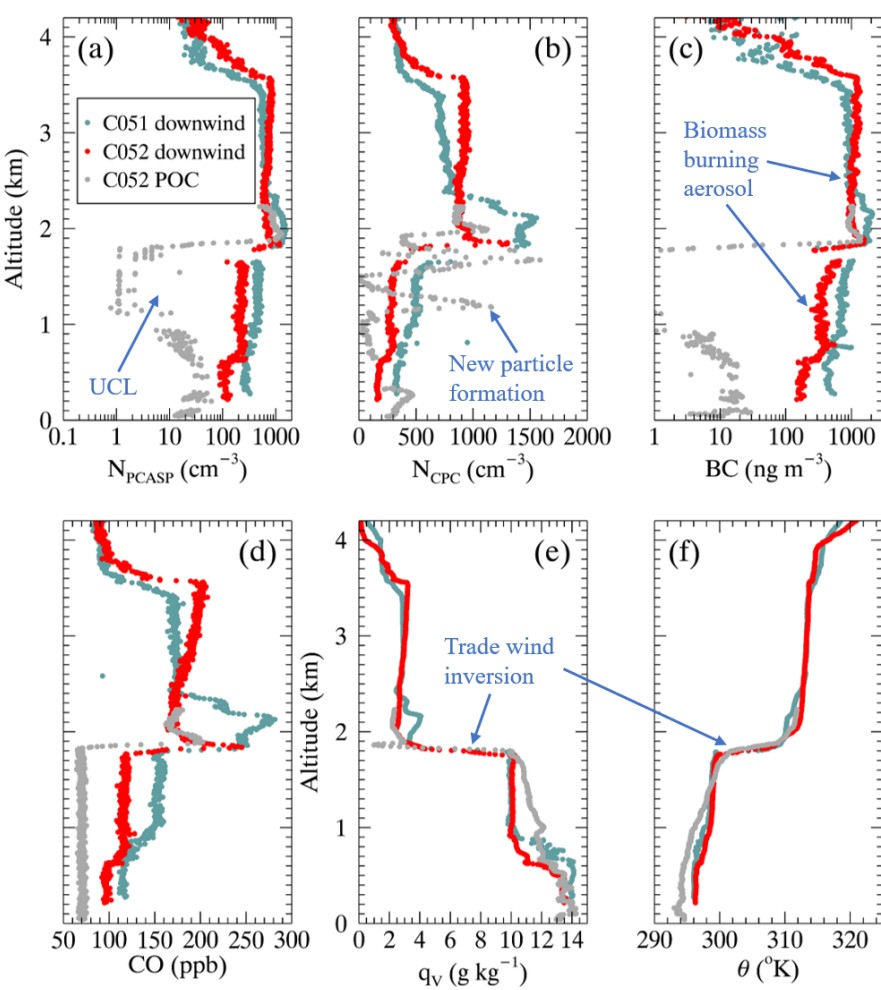

**Figure 5.** Vertical profiles of PCASP number concentration, CPC number concentration, BC mass concentration, carbon monoxide, water vapor mixing ratio and potential temperature within and downwind of the POC.





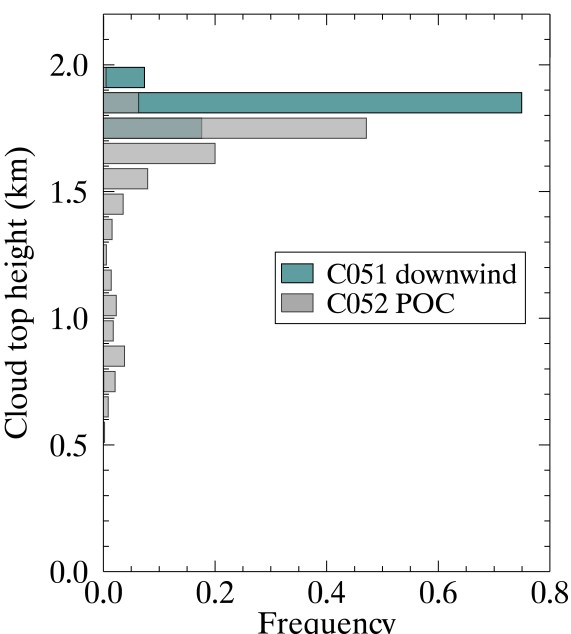

**Figure 6.** Vertical profiles of the frequency of cloud top height measured from the aircraft lidar when flying above the boundary layer. Data measured over the POC on flight C052 and over the closed cells downwind of the POC on flight C051 are shown.





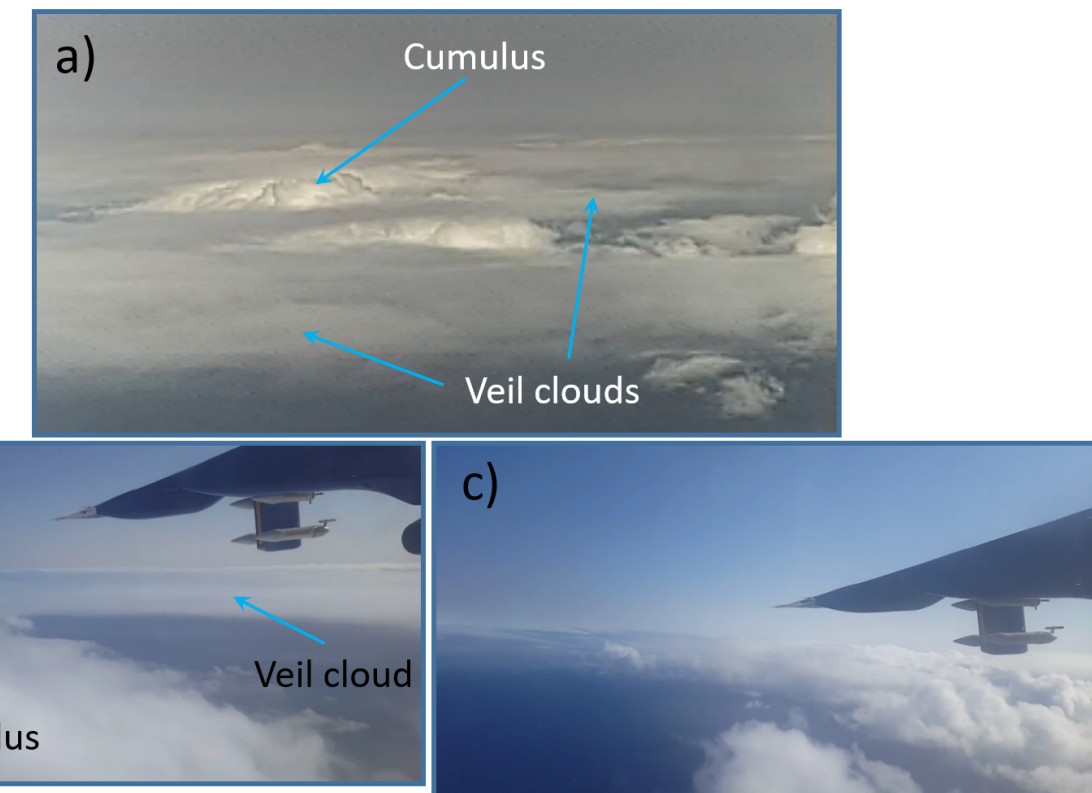

**Figure 7.** Photographs taken from the FAAM aircraft on flight C052. Photo a) is from the aircraft rearward facing camera on profile P1 at 15:44 UTC, as the aircraft headed westwards on the initial descent into the POC. This was just above the boundary layer at an altitude of 2286 m and at 8.3° W. Photos b) and c) are from the return low-level leg. b) was taken at 16:12 UTC, 10.2° W, at an altitude of 1847 m as the aircraft climbed above the boundary layer on profile P4. c) was taken at 16:25 UTC, 11.0° W, within the ultra-clean layer on profile P6 at an altitude of 1240 m. This is where the enhanced aerosol concentration measured on the CPC was observed in the UCL (see Fig 5b). Photographs b) and c) are courtesy of Ross Herbert.

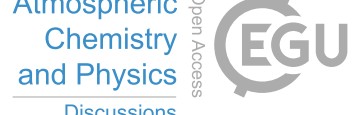



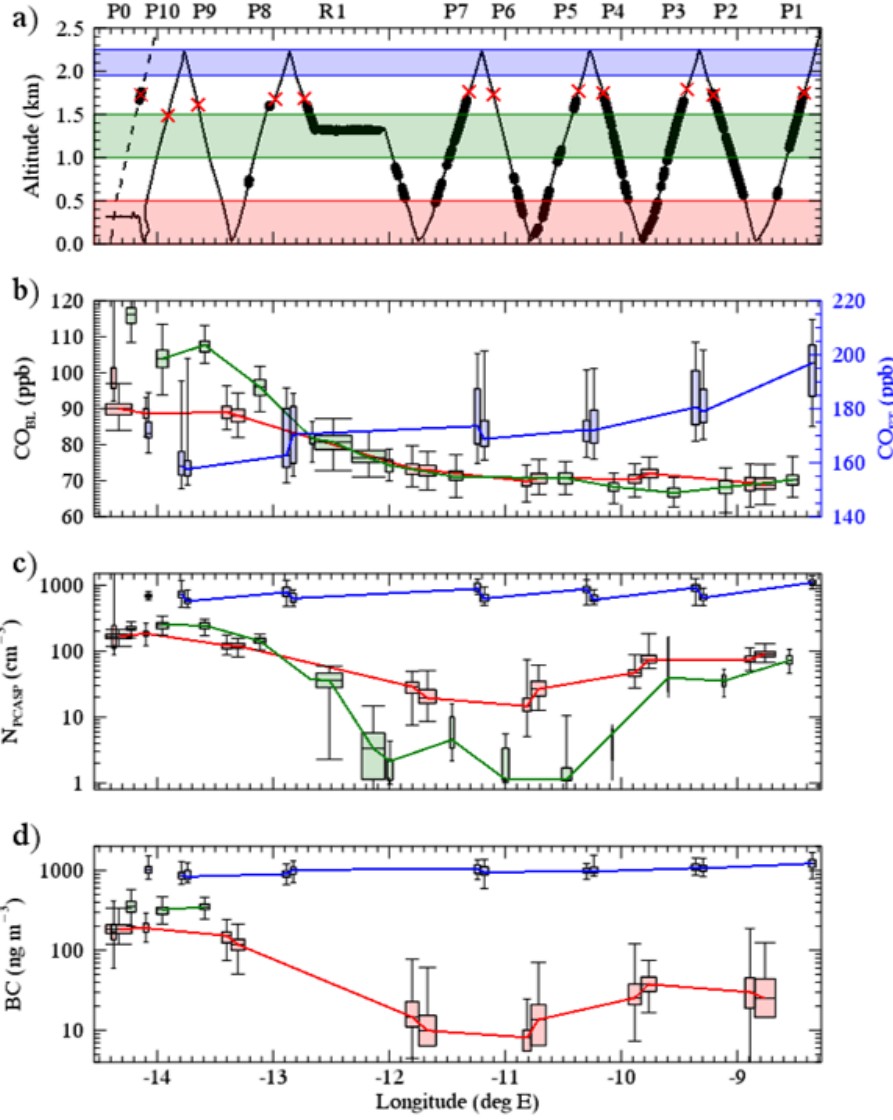

**Figure 8.** Aircraft data from flight C052 plotted as a function of longitude. Panel a) shows the aircraft altitude on the initial profile out of Ascension Island (dash line) and on the low-level return leg through the POC feature (solid line). The black filled circles overlaid on the flight track indicate where cloud or precipitation was sampled. The red crosses show the altitude of the base of the trade-wind inversion. The shaded regions indicate the altitude ranges used to calculate the data presented in panels b) to d). These correspond to data from the surface mixed layer (red), the ultra-clean layer (green) and the free-troposphere (blue). Panels b) to d) present along track measurements of CO, PCASP number concentration and BC mass concentration as box and whisker plots in the three altitude segments. Note that BC mass concentrations in the UCL (green) were below the detectable limit east of 12° W and that the free-tropospheric CO data in panel b) uses the right-hand y-axis. Profile and run names are indicated on the top of the figure.

**Figure 9.** Panels a) and b) show SEVIRI RGB imagery at 10 and 16 UTC on the 5 September 2017. Aircraft tracks are shown with a thin white line from flight C051 (10 UTC image) and flight C052 (16 UTC image). The position where aircraft profiles (P0 to P10) and run (R1) from flight C052 were made are shown with a star on both images. The position of these points at the time of each satellite image as calculated from trajectories initialised at each measurement location are shown with open circles. Panels c) and d) show MODIS true color imagery with the 3.7 $\mu$m effective radius retrieval overlaid. The imagery in c) and d) was obtained from NASA Worldview. The overpass times are indicated at the top of each figure.



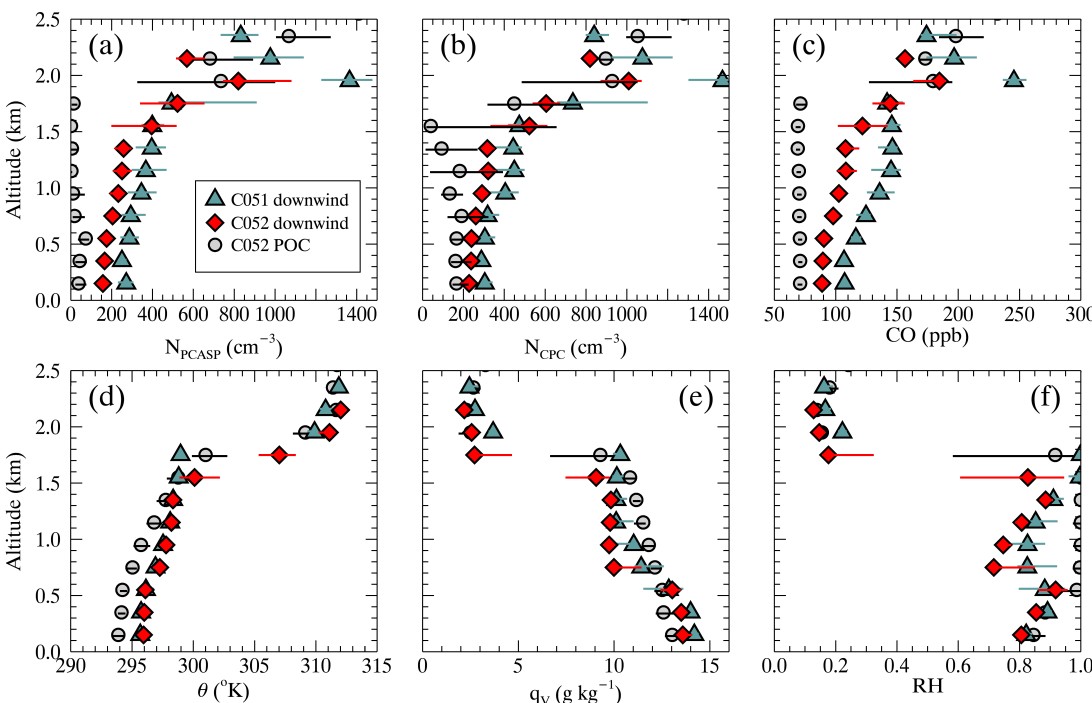

**Figure 10.** Composite vertical profiles of a) PCASP number concentration b) CPC number concentration c) carbon monoxide d) potential temperature e) specific humidity and f) relative humidity. The filled symbols are the median and the horizontal bars the inter-quartile range of the data. For flight C052, the data points are from the return low-level leg. The POC is defined as points east of 12° W and the downwind airmass data to the west of 13° W.



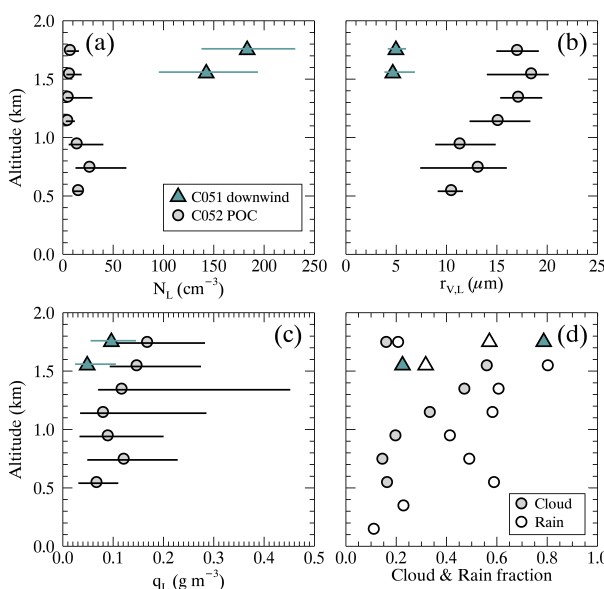

**Figure 11.** Composite vertical profiles of a) cloud drop number concentration b) volume mean radius of cloud drops and c) cloud liquid water content. The filled symbols are the median and the horizontal bars the inter-quartile range of the in-cloud data. Panel d) shows vertical profiles of the cloud (filled symbols) and rain (open symbols) fraction from the measurements along the flight track. Cloudy points are defined when the CDP LWC $> 0.01$ g m$^{-3}$ and rain points when the concentration of drops larger than 60 $\mu$m exceeds 1 L$^{-1}$.



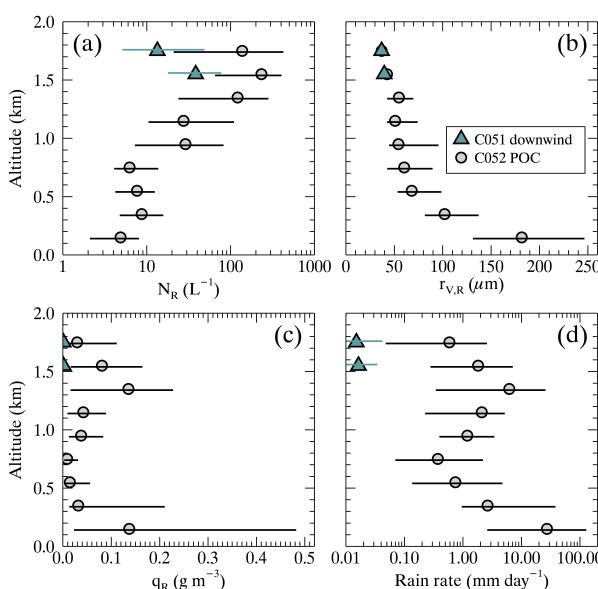

**Figure 12.** Composite vertical profiles of a) rain drop number concentration b) volume mean radius of rain drops c) rain water content and d) rain rate. The filled symbols are the median and the horizontal bars the inter-quartile range of the in-rain data. Data points containing rain are defined when the concentration of drops larger than 60 $\mu$m exceeds 1 L$^{-1}$.



**Figure 13.** Aircraft observations from run R1 on flight C052 at 1320 m altitude covering a distance of ∼ 70 km. The panels show a) vertical velocity; b) liquid water content calculated from the CDP, Nevzorov TWC sensor and integrating the composite particle size distribution; c) number concentration of cloud drops (CDP), rain drops ($N_R$) and out of cloud accumulation mode aerosol number concentration (PCASP); d) the effective radius calculated from the CDP and the composite PSD; e) precipitation rate from the composite PSD. Panels f) and g) contrast the mean size distributions averaged over an active Cu cell and a more quiescent cloud at the times indicated by the grey shading. The mean in-cloud size distribution from flight C051 is shown with a dashed line for comparison.

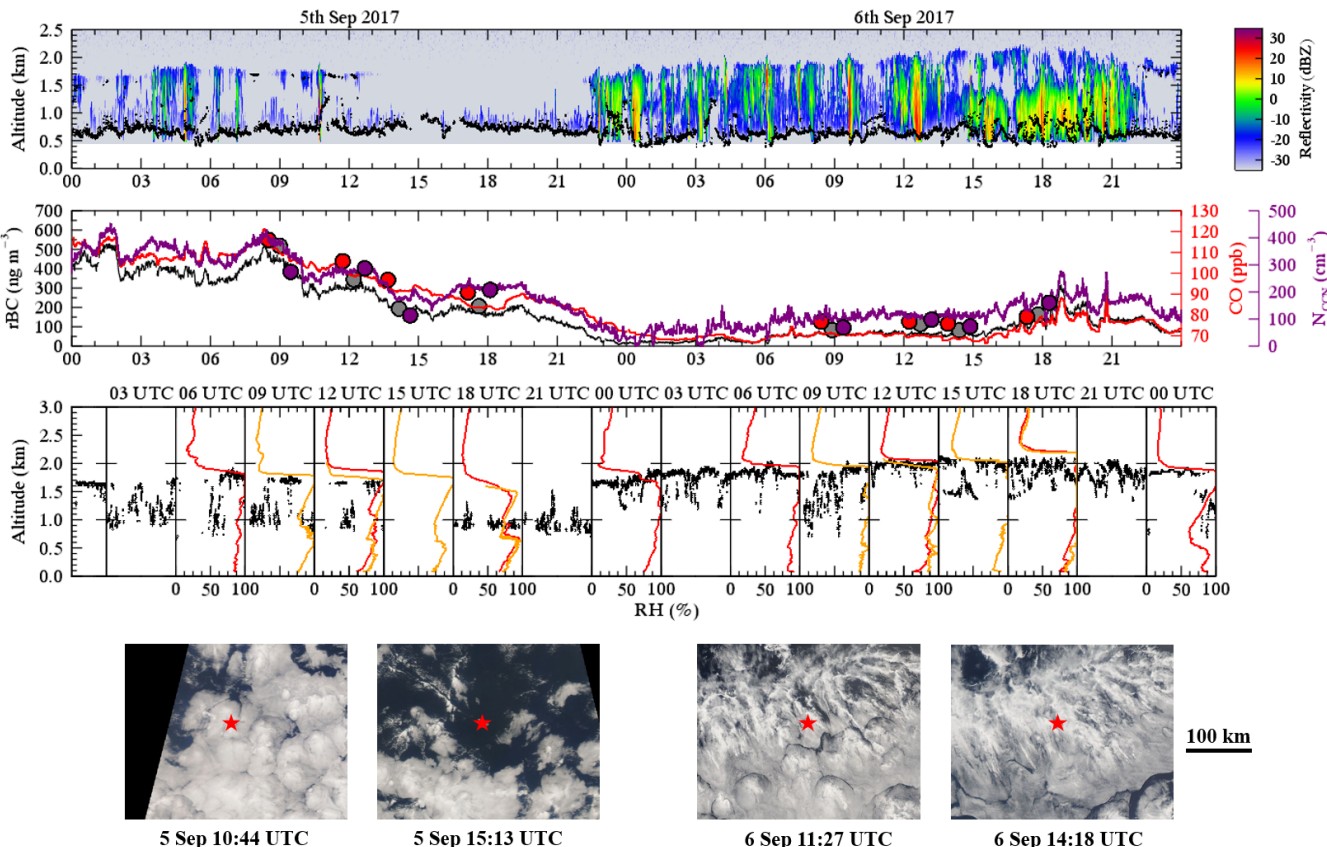

**Figure 14.** Observations from Ascension Island on the 5th and 6th September 2017. The top panel shows radar reflectivity from the Ka-band zenith radar. Cloud base height measurements from the ceilometer are overlaid in black. The middle panel shows measurements of the BC mass concentration, carbon monoxide and CCN concentration, made at the LASIC ARM site. Overlaid with filled circles are measurements from the FAAM aircraft made below 500 m altitude during profiles in and out of Ascension Island. The aircraft PCASP concentration is used as a proxy for the CCN concentration. The third panel shows 3 hourly snapshots through the period. The black points are cloud top height estimates from the Ka-band radar taken from a ± 0.5 hr window around each 3 hour point. Overlaid are radiosonde (red) and aircraft profiles (orange) of relative humidity. MODIS true-color satellite imagery around Ascension Island (red star) are also shown when available. The MODIS imagery was obtained from NASA Worldview.



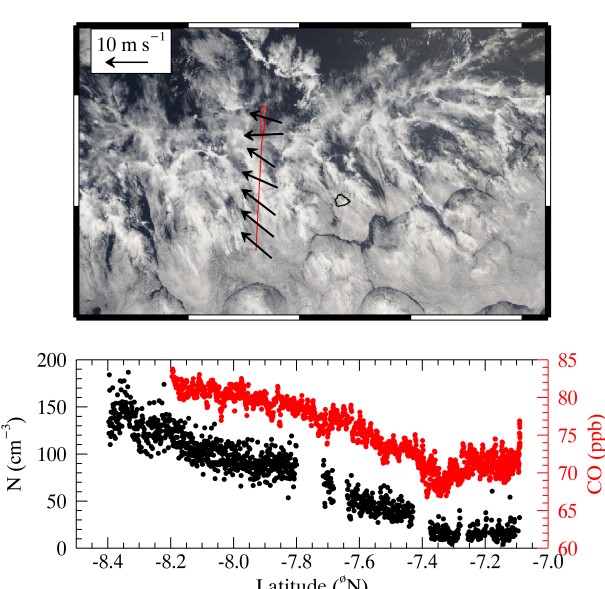

**Figure 15.** PCASP number concentration and carbon monoxide data on the 6th September 2017 (10:06 to 10:34 UTC) measured on flight C053. The data are to the west of Ascension Island and in the surface mixed layer (314 m altitude). The run crossed the boundary between open cells to the north and more stratiform cloud to the south. The flight track and measured wind vectors are overlaid on a MODIS Terra image valid at 11:27 UTC. The MODIS image was obtained from NASA Worldview.



**Figure 16.** Evaluation of the open cell fraction from 19 years of MODIS Terra imagery. Panel a) and b) are illustrative examples that show the manual identification of open cell features in red. The MODIS imagery was obtained from NASA Worldview. Panel c) shows a spatial map of the September 2000-2018 mean open cell fraction (0.02, 0.05, 0.1, 0.2 and 0.25 contours). The grey shaded region represents the area where the mean fine mode aerosol optical depth exceeds 0.2 and the black dashed line indicates the 0.6 total liquid cloud fraction contour. We approximate the area where both the September 2000-2018 mean fine mode AOD > 0.2 and the cloud fraction > 0.6 with the light blue shaded region. The blue line in panel d) shows the daily open cell fraction and the corresponding open cell area calculated in this blue shaded region for September 2000-2018. The box and whiskers summarize the individual monthly data. The open cell fraction for the example cases in panels a) and b) in the blue shaded region is 0.43 and 0.10.

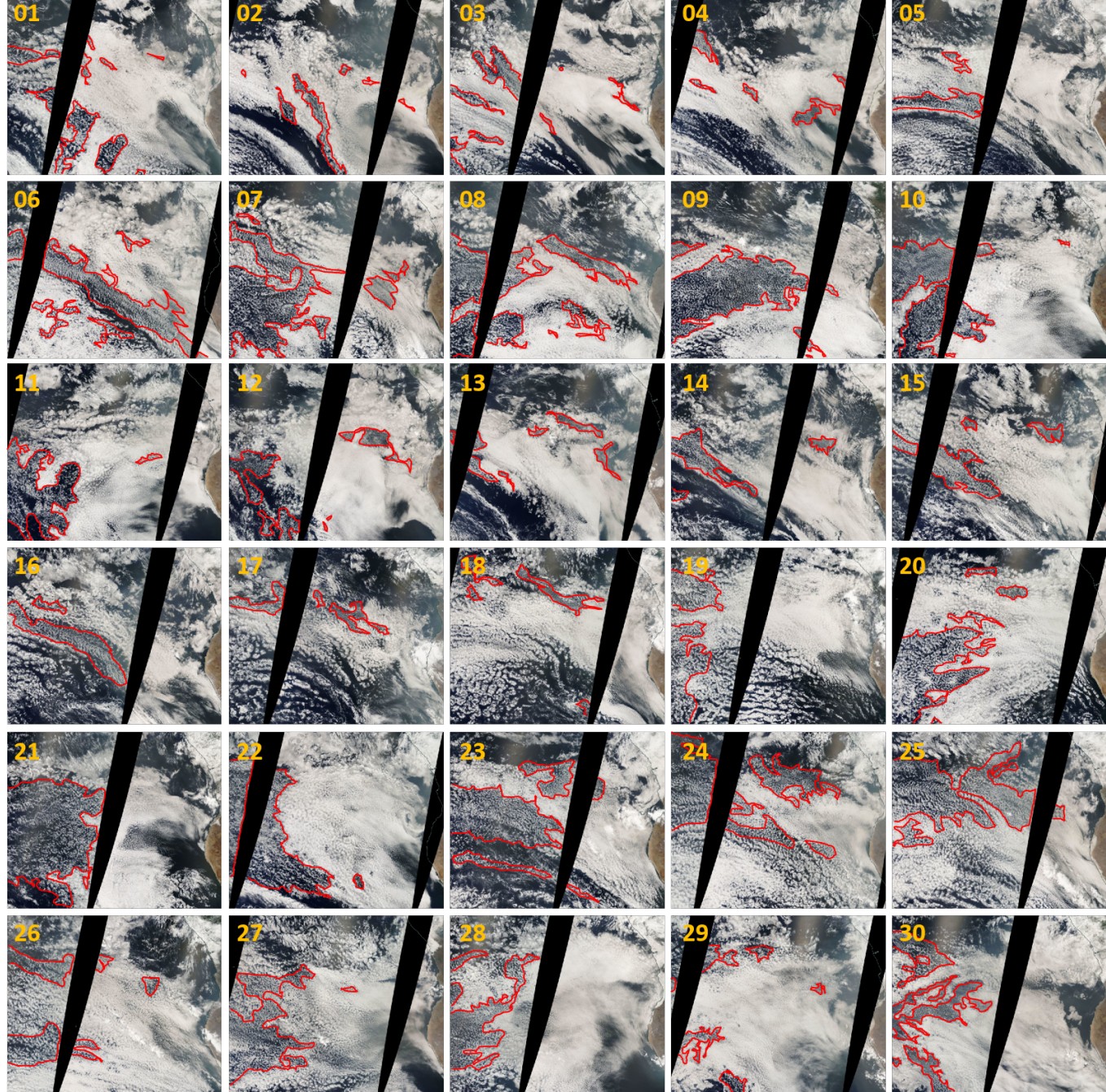

**Figure 17.** The manual identification of open cell regions in day-time MODIS Terra imagery are shown in red for the 1st to 30th September 2010. The MODIS imagery was obtained from NASA Worldview.