# Peer review of "Open cells can decrease the mixing of free-tropospheric biomass burning aerosol into the south-east Atlantic boundary layer"

_Atmospheric Chemistry and Physics, 2019_

## Referee Comment (RC1) · Anonymous Referee #1 · 2 Nov 2019

This manuscript reports observations that were obtained over the south Atlantic ocean. It documents closed and open cells under the influence of free tropospheric (FT) biomass burning aerosols. The main conclusion is that the overlying biomass burning aerosol are mixed more efficiently into the cloud layer in the overcast regions, while in the POC mixing is largely reduced. This has implications regarding aerosol-cloud interaction assessments. The study combines nicely in-situ observations with satellite and back-trajectories analysis. The manuscript is well written, though should be shorten and re-organized in some parts.

Major comments: The key results is based on aircraft observations of FT aerosols

above and within the boundary layer. Even though that based on the presented observational data the authors arguments are convincing, I'm still questioning whether it not solely rainout and cloud cleansing processes that are responsible for the cleaner POC (a question of time-scales of mixing vs rainout). I therefore expect a more comprehensive physical discussion regarding the reason why entrainment of FT aerosol into the boundary layer is less efficient in POC. This is needed to strengthen the authors conclusion.

Many of the observational findings in this study are in agreement with previous observational studies. Given that the aim of this study is not reporting observations (as far as I understand), I would expect the text to be more concise and focused on observations that are relevant to the main point of the study, rather than reporting many (perhaps less relevant) observational details.

Along the comment above, the manuscript is too descriptive in my view. I recommend it to be shorten. Also, methodological details are given within the result sections. These parts should be moved to the Methodology section.

Minor comments:

Can you use CALIOP data to complement the aircraft analysis regarding the aerosol layer height with respect to the clouds?

P2 L 24: https://doi.org/10.1002/2015GL066544 may be relevant to this discussion.

P2 L 34: https://doi.org/10.5194/acp-15-7351-2015 have shown that aerosol do able to close open cells. This is supported by observations: https://doi.org/10.1029/2012JD017981.

P4 L1: Entrainment should get more focus in the discussion here.

P5 L15: Which data product was used?

P6 L19: Mention here and later that these are Back-trajectories.

[Figure]

P6 L31: What is the size of the region over which the cloud fraction is estimated?

P7 L8: Please provide reference.

P7 L9-10: Aircraft data at cloud top? Otherwise it cannot be compared to satellite observations.

P8 L18: Sentence is not clear.

P8 section 4: Many parts in the paragraph starts here should be moved to the Method Section.

P10 L1-2: Please provide reference.

P11 L4: Not efficient in comparison to cleaning due to rainout.

P11 P26: What is the mechanism? Why open cells mixes FT air less efficiently?

P12 L12: Parts here can move to the Method Section.

P12 L15: Why lower inversion is associated with cloud clearing?

P13 L30- : Can be shorten. The caption should provide this information.

P15 L20: It's hard to see differences in Fig 11d with the current color scale (especially after printing).

P16: How these observations are relevant to the aim of the current study? Precipitation in POCs were shown in many studies in the past.

P16 L22: Figure 13 covers quite a large region. How come there is only one precip-itating cell (active cu)? In open cells one can expect active cells every few tens of km.

P16 L24: remove space in 13a.

P17 L 21: shouldn't it be 0.1 cm-3 for quiescent clouds and 1cm-3 for the cumulus?

P18 L26: It is not convincing without measurements above the inversion.

P18 L32: From MODIS true color images there are no POC over the island.

P19 L23-: Why not having this in the Method section?

P20 L17: Climatological AOD is discussed here. AOD is available only when there are no clouds. So how can you relate high AOD with overcast conditions? What about co-variability between biomass burning and Sc regime? You mentioned earlier in the text that some open cell structure are transport northward from southern latitudes- I would assume they are transported also with a cleaner air mass above the inversion.

P20 L25: Should be in the Methods section.

P20 L30: It should be mention that the assessment you provide here is not based on causality (seecomment above regarding co-variability)

---

## Referee Comment (RC2) · Anonymous Referee #2 · 8 Nov 2019

Abel et al. (2019) present an observational case study focused on a pocket of open cellular (POC) convection that formed under a biomass burning aerosol layer in the southeast Atlantic. They utilize a range of aircraft observations from several flights in the CLARIFY field project, as well as ground observations from an ARM Mobile Facility deployed to Ascension Island for the LASIC project, back trajectories and satellite retrievals of aerosol and cloud properties around their area of study. The authors nicely synthesize this wide range of datasets. They conclude that the POC must have reduced entrainment rates relative to a nearby area of mostly stratiform cloud cover. This result would have important implications for aerosol-cloud interactions, especially over the southeast Atlantic.

[Figure]

One of my main comments is primarily editorial/organizational. There were several examples – notably Section 3, Section 5 and Section 9 – of a mismatch between the ordering of the figures/subfigures and presentation of results in the text. For example, I think it helps make the paper as clear as possible if you don't have to jump between looking at Figures 5e and 5f while reading the analysis, then go to Figure 6, which is a totally new type of result and is mostly disconnected, then go back to Figure 5a etc. . ., to continue with those results. For the most part, I found the analysis of individual sub-figures and results easy to follow. This would just be to make sure each section and the full figures are organized and presented sequentially.

The primary conclusion of the paper is based on the comparison of the observations in the POC region to the downwind stratiform region. The trajectory analysis seems to indicate that prior to the formation of the POC, that airmass was unlikely to have been entraining smoky air from above given the separation between the aerosol layer and cloud top seen in the CALIPSO curtain. It then moves into a region where the base of the aerosol layer is lower and there could be entrainment of particles into the MBL. One point that I think would be good to make very explicit is that the same history holds for the region that later gets sampled as the downwind stratiform area. I think indicating this point specifically for both sampling regions, rather than just for the POC, would strengthen your argument about differences in entrainment rates.

I also think more specific discussion of connection to prior results (a handful of papers are already cited) around differences in entrainment rates in and around POCs would bolster the paper.

A few other minor comments:

Page 2, Line 14: "lead to a reduction in the outgoing flux at the top of the atmosphere. . .". Just clarify which flux (e.g. shortwave, net radiative etc. . .)

Page 3, Lines 18 – 19: ". . .there being no in-situ measurements of these events to date." I think that phrasing could be more specific that you are pointing out there have

been no in-situ observations of the formation of a POC, not a POC generally.

Figure 2: A legend in the figure marking what the color of the curves mean would be helpful for quicker interpretation.

Figures 5d,e,f: Do you have any thoughts on why CO would be well mixed under the POC even though the thermodynamic profiles indicate decoupling?

Figure 6: I'm not clear what Figure 6 brings specifically to this analysis since it mostly just generally confirms other prior observations. Is there a direct connection to the entrainment rate argument?

Page 10, Line 34: You use the word 'pristine' to refer to aerosol concentrations of 1 – 2 cm-3 in the ultra-clean layer of the upper boundary layer, but it seems pretty likely that even the slightly higher 28 cm-3 closer to the surface would also be pristine (if referring to the absence of continental or anthropogenic influence).

Figure 17: I'm not sure this figure was particularly necessary as a main figure. Perhaps this could be included in a supplement if needed.
* * *

---

## Referee Comment (RC3) · Anonymous Referee #3 · 10 Nov 2019

In this case study Abel et al. analyzed observational data of aerosols, trace gases, and clouds collected in the southeast Atlantic during field campaigns to describe the evolution of a POC and its interaction with the aerosol layer sitting above. They found that boundary layer within the POC area was very clean with a large vertical gradient near the trade inversion, across which the accumulation mode aerosol increased by orders of magnitude, while in the downwind area of the POC observational evidence shows a strong entrainment of the overlying biomass burning aerosol into the closed-cell boundary layer. They conclude that the entrainment is very weak within the POC. They further developed a 19-year monthly (September) climatology of POC occurrence in the southeast Atlantic, which suggests a high possibility of biomass burning aerosol

in "contact" with POCs but without interactions. The authors also pointed out that the assumption of these overlying aerosols modulating the clouds is problematic and might be incorrectly represented in large-scale models that are incapable of simulating POCs and cloud-top entrainment.

The data and findings are novel and interesting. The paper is generally well written, although some sections, especially the results sections 3-8 can be better streamlined. Often times I got lost in the details, for example, going back and forth between different figures far away from each other. Below are specific comments.

1. The title statement is inaccurate and even a little misleading. The entrainment mixing is weaker in the POC area than in the surrounding closed-cell areas, but I don't think there is evidence in the paper that shows how the open cells "decrease" the mixing or entrainment. I suggest revising the title to reflect the main claim of the paper.

2. P2, L14-27: the use of "negative" or "positive" with direct, semi-direct and indirect effect is quite confusing. Maybe replace it with "cooling" or "warming".

3. P3, L21: the year of reference Savic-Jovcic and Stevens (2018) is incorrect.

4. P4, L31: change "vertical wind" to "vertical velocity"

5. P11, L24: Remove hyphen in "pre-cursor"

6. P11, L27: Again, is there evidence to support the important role of open-cell clouds controlling the mixing? I understand that the strong entrainment mixing in close cells can be driven by cloud-top radiative cooling, which is much weaker in open cells because of the low cloud fraction. The strong precipitation in open cells may stabilize the boundary layer. What else in open cells can control the mixing?

7. P12 and P13: The description of Figure 8 and Figure 9 is an example of giving too much detail. Much of the information can go to the figure captions and there is no need to repeat it in the main text.

8. P14, L1: Fig. 9c: are these high values for drizzle drops? They look too large for cloud droplets. 9. P15, L26: Is the number of 60 micron for rain drop radius of diameter?

10. P17, L18: Panel c of Fig. 13 doesn't seem to be aligned with the other panels.

11. P17, L32-34: Is there observation showing that the thin/quiescent clouds can last for hours and rain at 10 mm d-1? LES model results seem to show that the active cumulus only last for minutes once they start raining but the resulting cold pool outflow can produce new active clouds in the previous quiescent cloud area.

12. P18, L31: What's the background level of BC and CO at the surface?

13. P19, L18: is "20 to 12%" correct?

14. P20, L5-9: the manual identification of POC is a little arbitrary. By looking at the examples in Fig. 17, it seems that the POC identification is quite conservative. Does that mean the 0.25 open cell fraction represent a lower bound? Also, in the formulation, were clear-sky and land surface pixels counted in P_ALL or P_BLACK?

---

## Author Comment (AC1) · 21 Dec 2019

We would like to thank all three reviewers for their constructive comments, that have helped to strengthen the manuscript. We note that there are two common themes picked up in the reviews, that we address collectively first. We then follow this by addressing the other specific comments from the individual reviewers in turn.

The two common themes and our response to these are as follows;

1. All three reviewers would like to see the text re-structured in places to avoid jumping back and forth between figures and to also look for opportunities to streamline the manuscript. We have therefore made the following changes:

   a. Moved section 4 that described the flight patterns into section 2.
   b. Moved some additional text that describes methodology into section 2.
   c. Either removed unnecessary text from the main body or moved text describing Figs. 1, 3 8, 9 of the submitted manuscript to the captions.
   d. Incorporated the figure that showed the cloud top height from the lidar (Fig. 6 of the submitted manuscript) into Fig 5. This reduces the number of figures and also enables the lidar data to be compared more easily to the thermodynamic profiles.
   e. Moved Fig 17 of the submitted manuscript to a supplement.
   f. We acknowledge that there were occasions where the reader was referred to figures later in the paper e.g. pointing to consistent features in the ground based data when discussing the aircraft observations and that this was not helpful. We have made efforts to ensure that the figures are now referred to in-sync in the revised version. We note that there are a few remaining occasions where we refer the reader back to a previous figure for comparison, but feel that this is appropriate.

2. To provide some additional discussion around the mechanisms as to why entrainment is lower in the open cell regions compared to closed cell regions. We address this by expanding our discussion in the conclusions section (blue italic text in the following paragraph).

   All of these features therefore suggest that the organized open cellular convection in the POC is very inefficient at entraining the overlying smoke into the marine boundary layer. The reduced efficiency in the mixing of free-tropospheric aerosols down into open cell boundary layers is consistent with previous inferences made from measurements of POCs in the south-east Atlantic (Wood et al. , 2011; Terai et al. , 2014) and from observations of the stratocumulus to cumulus transition in cold-air outbreaks (Abel et al. , 2017). We note however that these former studies have exhibited significantly lower free-tropospheric accumulation mode aerosol concentrations in contact with the inversion above the UCL ($\sim$ 10 to 100 cm $-3$ ) than were observed from the measurements on this case-study. *A weaker entrainment rate across the boundary layer inversion into the POC is also consistent with previous cloud-resolving model studies. For example, simulations of trade wind cumulus capped by a strong inversion have demonstrated that entrainment rates and cloud fraction are tightly coupled, with increased stratiform cloud cover promoting more mixing across the inversion through enhancements in turbulence generated from cloud top radiative cooling (Stevens et al. ,2001; Lock, 2009). This* is also consistent with arguments made by Bretherton et al. (2010) and latter cloud-resolving model studies of POCs that demonstrate a much weaker entrainment rate within the open cells compared to the modelled surrounding overcast cloud field (Berner et al. ,2011, 2013). *Although the observations in this case-study show the presence of thin stratiform veil clouds within the*

*POC, these are shown to exhibit low levels of turbulence, in accordance with previous measurements of these cloud features in open cell regions (Wood et al. 2018) and so would be expected to contribute weakly to entrainment. Whilst the intermittent active cumulus turrets in the POC could penetrate across the strong trade-wind inversion and locally mix down free-tropospheric biomass burning aerosol into the boundary layer, the prevalence of low CO values in the UCL suggests that this mixing does not dominate the aerosol budget of the POC. In contrast, the measurements of a more polluted boundary layer in the overcast stratiform region surrounding the POC are consistent with that cloud generating stronger and more widespread mixing across the inversion.*

We now address the specific comments from each of the three reviewers in turn. Reviewer comments are in black text and our response in blue text.

**Reviewer 1**

This manuscript reports observations that were obtained over the south Atlantic ocean. It documents closed and open cells under the influence of free tropospheric (FT) biomass burning aerosols. The main conclusion is that the overlying biomass burning aerosol are mixed more efficiently into the cloud layer in the overcast regions, while in the POC mixing is largely reduced. This has implications regarding aerosol-cloud interaction assessments. The study combines nicely in-situ observations with satellite and back-trajectories analysis. The manuscript is well written, though should be shorten and re-organized in some parts.

Major comments: The key results is based on aircraft observations of FT aerosols above and within the boundary layer. Even though that based on the presented observational data the authors arguments are convincing, I'm still questioning whether it not solely rainout and cloud cleansing processes that are responsible for the cleaner POC (a question of time-scales of mixing vs rainout).

**The key observation that demonstrates that entrainment of the overlying aerosol is weaker in the POC comes from the carbon monoxide (CO) measurements. CO can be used as a tracer of the overlying biomass burning aerosol airmass as it is not readily removed by cloud and precipitation processes. The observations in the POC show clean background values of CO and this increases across the boundary to the airmass that contains the closed cells. If entrainment rates were similar in both cloud regimes and that it was solely rainout and cloud cleansing processes that resulted in the cleaner POC, we would have expected to see elevated CO in the POC. We had stated this in both the abstract and conclusions section. We do include some additional text in the revised version to further emphasize this.**

I therefore expect a more comprehensive physical discussion regarding the reason why entrainment of FT aerosol into the boundary layer is less efficient in POC. This is needed to strengthen the authors conclusion.

**We have expanded the text on why entrainment may be expected to be lower in the open cell region (based on prior studies) in the conclusions and discussion section. Please see the collective response to all three reviewers for further information.**

Many of the observational findings in this study are in agreement with previous observational studies. Given that the aim of this study is not reporting observations (as far as I understand), I would expect the text to be more concise and focused on observations that are relevant to the main point of the study, rather than reporting many (perhaps less relevant) observational details. Along the comment above, the manuscript is too descriptive in my view. I recommend it to be shorten. Also, methodological details are given within the result sections. These parts should be moved to the Methodology section.

**Although some of the observations presented in this study are in agreement with previous work, we feel that it is worthwhile to report them here as i) airborne measurements of POCs in the literature are still sparse and ii) the observations presented will be useful for evaluating LES simulations of this case that are underway. That said, we have made efforts to streamline and re-organize the text. Please see the collective response to all three reviewers for further information.**

Minor comments:

Can you use CALIOP data to complement the aircraft analysis regarding the aerosol layer height with respect to the clouds?

**We had included the CALIOP and CATS feature mask from overpasses that were relatively close in time and space to the POC airmass over the preceding 5 days. These were shown in Fig 3 and discussed in section 3 of the submitted manuscript. They support the idea that the overlying aerosol layer was unlikely to have been in contact with the clouds when the POC formed. As the POC airmass travelled northwards, the spaceborne lidars indicate that the base of the overlying aerosol layer descended and contact with the clouds was likely to have been made north of about 15 °S.  Given the limited spatial and temporal coverage of CALIOP and CATS, there were no other suitable overpasses in the vicinity of the measurements for this case-study.**

P2 L 24: https://doi.org/10.1002/2015GL066544 may be relevant to this discussion.

**We now include some more discussion in the introduction and refer to the Yamaguchi et al. paper to highlight i) the subtleties found in models for the sign and magnitude of the semi-direct effect when an absorbing aerosol layer overlays cloud and ii) additional model evidence for the role of entrained smoke in brightening the cloud field through microphysical interactions.  The updated sections of the text are**

*The result is often to increase the amount of cloud condensate and brighten the stratocumulus, resulting in a net cooling of the atmosphere (a negative semi-direct effect) e.g. Johnson et al. (2004);Wilcox. (2010). However, model studies demonstrate that both the sign and magnitude of this cloud response is highly sensitive to a multitude of factors, including the properties of the overlying aerosol layer and the thermodynamic structure of the boundary layer (Yamaguchi et al. , 2015; Herbert et al. , 2019).*

*Once these aerosols have been entrained into the boundary layer they can provide an additional source of cloud condensation nuclei (CCN), that can then result in a modification of the cloud properties (Diamond et al. , 2018) and a brightening of the cloud field, increasing the shortwave flux reflected back to space (a negative indirect effect) e.g. Yamaguchi et al. (2015); Lu et al. (2018).*

P2 L 34: https://doi.org/10.5194/acp-15-7351-2015 have shown that aerosol do able to close open cells. This is supported by observations: https://doi.org/10.1029/2012JD017981.
**Thank you for pointing out these articles – they are indeed relevant to the discussion on the potential role that increased aerosol can play in transitioning open cells to closed cells. We have modified the text slightly later in the introduction (P3 L34) to incorporate these. The modified text now reads.**

*A question then arises as to how the cloud in a well-developed POC may respond to a large CCN perturbation from entrained biomass burning aerosols. Visual evidence form satellite imagery demonstrates that in nature, the transition from open cells to a more overcast state can occur when the boundary layer is exposed to large injections of CCN from surface based ship traffic e.g. Goren and Rosenfeld , 2012. The idealised model study of Wang and Feingold (2009b) also shows that an abrupt change in the CCN concentration in a mature POC can shut off precipitation and lead to cloud fraction increases, although this is not sufficient for the open cells to fully transition to the closed cell state in that case. That said, even a moderate change in cloud fraction could have important consequences for the direct and indirect effects in the region. However, the model studies of Berner et al. (2011, 2013) suggest that entrainment may be much weaker in POCs, due to a reduction in the amount of turbulence generated at cloud top when compared to the surrounding overcast stratocumulus cloud field, which could limit how readily overlying biomass burning aerosol can be entrained into a POC. This reduced entrainment could therefore serve as an effective barrier to large and rapid perturbations of CCN, limiting the ability of these open cells regions to transition to closed cells (Feingold et al. , 2015). A key focus of this work therefore*

*examines if there is observational evidence of differences between how subsiding biomass burning aerosol plumes are mixed down into the measured POC and surrounding overcast cloud regimes.*

P4 L1: Entrainment should get more focus in the discussion here.

**We have modified the text (see response to above comment P2 L34) to now state that the reduced entrainment in the POC is *"due to a reduction in the amount of turbulence generated at cloud top".* We also expand out discussion on entrainment in the discussion section – see collective response to all reviewers.**

P5 L15: Which data product was used?

**We are a little unclear as to which satellite data/imagery the reviewer is referring to here. The SEVIRI RGB and IR imagery was generated by the lead author. The IR imagery uses the 10.8 μm brightness temperature (as noted in the Fig 1 caption) and the RGB imagery uses a combination of the 0.6, 0.8 and 1.6 μm channels. The individual SEVIRI channel data were obtained from EUMETSAT and we have added an acknowledgement. The MODIS imagery was from NASA Worldview as noted in the relevant figure captions, but we now also include that statement in this section. The MODIS effective radius imagery included in Figure 9 was also from NASA worldview. Although we had referenced Platnick et al. (2017a), we now explicitly state that this is level 2 collection 6 data. All other satellite data products are referenced.**

P6 L19: Mention here and later that these are Back-trajectories.

**Backward and forward trajectories are used in the paper. Whilst most of the trajectory data in the manuscript uses back trajectories, the position of the open blue circle at the time of the satellite image in figure 1f for example is from a forward trajectory. The use of both backward and forward trajectories was mentioned in the methods section (section 2.4).**

P6 L31: What is the size of the region over which the cloud fraction is estimated?

**All satellite data (including cloud fraction) presented in Fig 2 was from a 1 x 1 degree latitude-longitude box, as was stated on line P6 line 25 and in the caption of the figure.**

P7 L8: Please provide reference.

**The SEVIRI retrieval of effective radius is that of Peers et al. (2019) that is described in the methods section. We have also added the reference at the beginning of this paragraph.**

P7 L9-10: Aircraft data at cloud top? Otherwise it cannot be compared to satellite observations.

**Yes, we agree. The data in the closed cell region were already averaged over the top 50 m of the cloud layer. In the open cell region, we had taken an average of data over the full depth of the clouds. This was because we recognised that the clouds in the open cell region are of varying depth as illustrated by the lidar measurements in figure 6 of the submitted manuscript. However, based on the reviewers comment, we now also restrict this to the top of the boundary layer (upper 100 m). This results in smaller values (reff = 23 μm rather than 29 μm). The smaller values towards the boundary layer top are consistent with the results shown in Fig 12b. We have updated figure 2 and added a sentence in the revised manuscript to highlight that the data are representative of the cloud top region.**

P8 L18: Sentence is not clear.

**We have re-phrased the section of text to make it clearer. It now reads**

*We also plot additional back trajectories that are started at 12 hourly intervals back along the black 500 m trajectory, in order to examine the time-history of where the free-tropospheric air that is entrained into the boundary layer originates from. The orange (T-12), dark blue (T-24), purple (T-36) and light blue (T-48) stars are the additional trajectory start points. The start height of each of these is adjusted to remain just beneath the boundary layer inversion, that lowers to the south.*

P8 section 4: Many parts in the paragraph starts here should be moved to the Method Section.

**We have moved the whole section to the methods section (section 4 of the submitted manuscript has become the new section 2.5).**

P10 L1-2: Please provide reference.

**We have referenced Stevens et al. (2001) and Lock (2009).**

P11 L4: Not efficient in comparison to cleaning due to rainout.

**We have moved the statement "suggesting that entrainment of overlying aerosol into the POC is not an efficient process" to the next paragraph, after the carbon monoxide data are described and where cleaning of aerosols by precipitation is discussed.**

P11 P26: What is the mechanism? Why open cells mixes FT air less efficiently?

**We have added some more discussion on possible reasons why open cells may be expected to entrain free-tropospheric air less efficiently in the conclusions section. Please see the collective response to all three reviewers for further information.**

P12 L12: Parts here can move to the Method Section.

**We have moved parts of this text to the figure caption.**

P12 L15: Why lower inversion is associated with cloud clearing?

**The lowering inversion is associated with a reduction in relative humidity at the top of the boundary layer (below saturation) as seen in Fig 10f and Fig 14. Although this was discussed later in the manuscript, we now also briefly mention it at this stage in the revision. The modified sentence reads**

*This is associated with a reduction in the relative humidity at the top of the boundary layer (below saturation) and the large-scale cloud clearance in the afternoon downwind of the POC.*

P13 L30- : Can be shorten. The caption should provide this information.

**We have moved parts of this text to the figure caption.**

P15 L20: It's hard to see differences in Fig 11d with the current color scale (especially after printing).

**We have revised the figure by linking the individual symbols with a line for both the cloud and rain fractions, to enable the differences to be seen more clearly.**

P16: How these observations are relevant to the aim of the current study? Precipitation in POCs were shown in many studies in the past.

**We also want to document the case as fully as possible for LES model evaluation of this case-study that is underway.**

P16 L22: Figure 13 covers quite a large region. How come there is only one precipitating cell (active cu)? In open cells one can expect active cells every few tens of km.

**We refer the reviewer to the spatial scales present in the open cell region in Fig 9b, with the aircraft flight track overlaid. Given that the aircraft is flying along a straight line through the cloud field, it is not unreasonable to expect that only one region of active Cu (bright cells in the image) would be sampled over a 70 km distance.**

P16 L24: remove space in 13a.

**Done**

P17 L 21: shouldn't it be 0.1 cm-3 for quiescent clouds and 1cm-3 for the cumulus?

**These values are the concentration of precipitation sized particles (green line in Fig 13c) and are correct.**

P18 L26: It is not convincing without measurements above the inversion.

**We agree and so have examined the data from the four flights that cover the time period shown in Fig 14 and now include a sentence in the revised manuscript stating that**

***Examination of the aircraft measurements in the free troposphere confirms that the base of the biomass burning aerosol layer remained in contact with the boundary layer inversion throughout the period (not shown).***

P18 L32: From MODIS true color images there are no POC over the island.

**We described the cloud conditions on P18 L23 of the submitted manuscript, which reads "*The associated cloud conditions can also be seen on the MODIS imagery from the 6th September in Fig. 14. The images suggest that the southern boundary of the remnants of the POC feature was roughly aligned west-east and located just to the south of Ascension Island.*" This is consistent with the evolution of the POC feature shown in Fig 1.**

P19 L23-: Why not having this in the Method section?

**We have chosen to leave the description of the open cell fraction calculation here, as at this stage there is a shift in the focus of the manuscript from the in-situ observations to the climatology of open cell fraction.**

P20 L17: Climatological AOD is discussed here. AOD is available only when there are no clouds. So how can you relate high AOD with overcast conditions? What about co-variability between biomass burning and Sc regime? You mentioned earlier in the text that some open cell structure are transport northward from southern latitudes- I would assume they are transported also with a cleaner air mass above the inversion.

**We do not try to infer any correlation between AOD and cloud conditions in this study. Rather, we simply use the climatological AOD and cloud cover to define our region of interest for the calculation of open cell fraction. We do state that given that the location of peak open cell fractions occurs is in a region of high AOD, "it is therefore plausible that subsiding free-tropospheric biomass burning aerosol layers transported from the continent may often come into contact with regions exhibiting open cell cloud morphologies". We do not think that this is an unreasonable statement to make. We have learnt from the CLARIFY field campaign for example, that free-tropospheric biomass burning aerosol plumes in contact with a cloud topped boundary**

**layer in the offshore environment are not uncommon at this time of year (Haywood et al. 2020 and Hui et al. 2020, both in preparation).**

**On the comment about transport pathways, the prevailing boundary layer flow in the stratocumulus region is south-easterly and so the direction of travel of the open cell structures that move northwards are not atypical. The transport of free-tropospheric biomass burning aerosol plumes from the continent out over the Ocean are in a separate airmass (until mixing occurs into the boundary layer). One cannot therefore relate a clean boundary layer airmass from southerly latitudes to aerosol conditions in the free-troposphere.**

P20 L25: Should be in the Methods section.

**The satellite datasets are described/referenced in the datasets section. We have moved this sentence to the figure caption.**

P20 L30: It should be mention that the assessment you provide here is not based on causality (see comment above regarding co-variability)

**See response to comment above. We did not mean to infer that this is based on causality and state that "it is plausible….."**

Reviewer 2

Abel et al. (2019) present an observational case study focused on a pocket of open cellular (POC) convection that formed under a biomass burning aerosol layer in the southeast Atlantic. They utilize a range of aircraft observations from several flights in the CLARIFY field project, as well as ground observations from an ARM Mobile Facility deployed to Ascension Island for the LASIC project, back trajectories and satellite retrievals of aerosol and cloud properties around their area of study. The authors nicely synthesize this wide range of datasets. They conclude that the POC must have reduced entrainment rates relative to a nearby area of mostly stratiform cloud cover. This result would have important implications for aerosol-cloud interactions, especially over the southeast Atlantic.

One of my main comments is primarily editorial/organizational. There were several examples – notably Section 3, Section 5 and Section 9 – of a mismatch between the ordering of the figures/subfigures and presentation of results in the text. For example, I think it helps make the paper as clear as possible if you don't have to jump between looking at Figures 5e and 5f while reading the analysis, then go to Figure 6, which is a totally new type of result and is mostly disconnected, then go back to Figure 5a etc. . ., to continue with those results. For the most part, I found the analysis of individual sub-figures and results easy to follow. This would just be to make sure each section and the full figures are organized and presented sequentially.

**We have made efforts to streamline and re-organize the text. With regards to Fig 5 and Fig 6, we have now combined these. Please see the collective response to all three reviewers for further information.**

The primary conclusion of the paper is based on the comparison of the observations in the POC region to the downwind stratiform region. The trajectory analysis seems to indicate that prior to the formation of the POC, that airmass was unlikely to have been entraining smoky air from above given the separation between the aerosol layer and cloud top seen in the CALIPSO curtain. It then moves into a region where the base of the aerosol layer is lower and there could be entrainment of particles into the MBL. One point that I think would be good to make very explicit is that the same history holds for the region that later gets sampled as the downwind stratiform area. I think indicating this point specifically for both sampling regions, rather than just for the POC, would strengthen your argument about differences in entrainment rates.

**The figure below is the same as Fig 3 a and b in the submitted version, but the trajectories are initialised from the flight in the closed cell region downwind of the POC. The trajectories show a similar airmass history and we have added the following comment in the manuscript.**

*"A comparable analysis with trajectories initialised in the stratiform cloud layer sampled immediately downwind of the POC show a very similar airmass history (not shown)."*

I also think more specific discussion of connection to prior results (a handful of papers are already cited) around differences in entrainment rates in and around POCs would bolster the paper.

**We agree and have expanded the conclusions and discussion section. Please see the collective response to all three reviewers for further information.**

[Figure]

A few other minor comments:

Page 2, Line 14: "lead to a reduction in the outgoing flux at the top of the atmosphere. . .". Just clarify which flux (e.g. shortwave, net radiative etc. . .)

**We have modified the sentence so that it now reads "*As the biomass burning aerosols can partially absorb sunlight, they can exert a net warming of the atmospheric column and lead to a reduction in the outgoing shortwave flux at the top of atmosphere…*"**

Page 3, Lines 18 – 19: ". . .there being no in-situ measurements of these events to date." I think that phrasing could be more specific that you are pointing out there have been no in-situ observations of the formation of a POC, not a POC generally.

**Yes, no in-situ observations of POC formation events is what we were trying to convey. This sentence has been modified and now states that there are "*no in-situ measurements of these formation events to date.*"**

Figure 2: A legend in the figure marking what the color of the curves mean would be helpful for quicker interpretation.

**We have added labels for the POC and stratiform cloud trajectories.**

Figures 5d,e,f: Do you have any thoughts on why CO would be well mixed under the POC even though the thermodynamic profiles indicate decoupling?

**We speculate that the boundary layer thermodynamic structure and CO was well mixed at some point before the POC formed. As the airmass moved northwards, the boundary layer would have advected over warmer SSTs and deepened, with heavy sustained drizzle developing in the POC,**

**both of which could contribute to the decoupled boundary layer structure. Given that there are no significant sources of CO in the boundary layer and that there was limited mixing of free-tropospheric CO into the POC airmass, then it is conceivable that the shape of the CO profile in the boundary layer could remain relatively invariant with height. However, given that we have no observations to demonstrate this, we have chosen to not include this in the revised manuscript. It might be something that is amenable to test in a LES simulation, with a tracer used for CO for example.**

Figure 6: I'm not clear what Figure 6 brings specifically to this analysis since it mostly just generally confirms other prior observations. Is there a direct connection to the entrainment rate argument?

**We would prefer to keep the lidar data in the paper, as it provides a more extensive spatial survey than is possible from the aircraft profiles alone. We have however incorporated Fig 6 into Fig 5 to reduce the figure numbers. A benefit is that is allows the reader to more easily compare the cloud top height frequencies against the illustrative thermodynamic profiles.**

**The similar cloud top height between the downwind closed cells and the POC does however support the conceptual model of Bretherton et al (2010) and the subsequent model simulations of Berner et al. (2011). These model studies suggest that despite the weaker entrainment in the open cell region, sharp horizontal gradients in inversion height cannot be supported across the POC boundary. To therefore compensate for the weaker entrainment in the open cells (that would otherwise act to significantly reduce inversion height), subsiding air would need to be channelled away from the POC i.e. subsidence rates are locally weaker above the open cells.**

Page 10, Line 34: You use the word 'pristine' to refer to aerosol concentrations of $1 - 2$ cm-3 in the ultra-clean layer of the upper boundary layer, but it seems pretty likely that even the slightly higher 28 cm-3 closer to the surface would also be pristine (if referring to the absence of continental or anthropogenic influence).

**We have kept the definition of pristine for the conditions found in the ultra-clean layer, but modify the sentence in the revised text to state that the low concentrations in the sub-cloud layer in the POC are still representative of unpolluted conditions.**

*"…..concentrations in the surface mixed layer are much lower than in the downwind profiles and representative of unpolluted conditions, with a mean value of 28 cm$^{-3}$."*

Figure 17: I'm not sure this figure was particularly necessary as a main figure. Perhaps this could be included in a supplement if needed.

**We have moved this to a supplement as suggested.**

Reviewer 3

In this case study Abel et al. analyzed observational data of aerosols, trace gases, and clouds collected in the southeast Atlantic during field campaigns to describe the evolution of a POC and its interaction with the aerosol layer sitting above. They found that boundary layer within the POC area was very clean with a large vertical gradient near the trade inversion, across which the accumulation mode aerosol increased by orders of magnitude, while in the downwind area of the POC observational evidence shows a strong entrainment of the overlying biomass burning aerosol into the closed-cell boundary layer. They conclude that the entrainment is very weak within the POC. They further developed a 19-year monthly (September) climatology of POC occurrence in the southeast Atlantic, which suggests a high possibility of biomass burning aerosol in "contact" with POCs but without interactions. The authors also pointed out that the assumption of these overlying aerosols modulating the clouds is problematic and might be incorrectly represented in large-scale models that are incapable of simulating POCs and cloud-top entrainment.

The data and findings are novel and interesting. The paper is generally well written, although some sections, especially the results sections 3-8 can be better streamlined. Often times I got lost in the details, for example, going back and forth between different figures far away from each other. Below are specific comments.

**We have made efforts to streamline and re-organize the text. Please see the collective response to all three reviewers for further information.**

1. The title statement is inaccurate and even a little misleading. The entrainment mixing is weaker in the POC area than in the surrounding closed-cell areas, but I don't think there is evidence in the paper that shows how the open cells "decrease" the mixing or entrainment. I suggest revising the title to reflect the main claim of the paper.

**Thank you for pointing this out. We have amended the title to "Open cells exhibit weaker entrainment of free-tropospheric biomass burning aerosol into the south-east Atlantic boundary layer"**

2. P2, L14-27: the use of "negative" or "positive" with direct, semi-direct and indirect effect is quite confusing. Maybe replace it with "cooling" or "warming".

**We have chosen to retain the use of the terminology "negative" and "positive" when referring to the radiative effects of aerosol-cloud-radiation interactions in the introduction, as this is standard notation in the literature. However, we do agree that cooling and warming, or referring to increased/reduced shortwave flux at TOA is easier for a general reader to understand and so we have added some additional wording to the text of the relevant sentences.**

3. P3, L21: the year of reference Savic-Jovcic and Stevens (2018) is incorrect.

**Thank you for spotting this typo. It has been corrected.**

4. P4, L31: change "vertical wind" to "vertical velocity"

**This has been changed to "Air vertical velocity".**

5. P11, L24: Remove hyphen in "pre-cursor"

**This has been changed as suggested.**

6. P11, L27: Again, is there evidence to support the important role of open-cell clouds controlling the mixing? I understand that the strong entrainment mixing in close cells can be driven by cloud-top radiative cooling, which is much weaker in open cells because of the low cloud fraction. The strong precipitation in open cells may stabilize the boundary layer. What else in open cells can control the mixing?

**We have added some more discussion on reasons why open cells may be expected to entrain free-tropospheric air less efficiently in the conclusions section. Please see the collective response to all three reviewers for further information.**

7. P12 and P13: The description of Figure 8 and Figure 9 is an example of giving too much detail. Much of the information can go to the figure captions and there is no need to repeat it in the main text.

**We have made efforts to streamline and re-organize the text. Please see the collective response to all three reviewers for further information. Some of the text describing Fig 8 and 9 has been added to the captions.**

8. P14, L1: Fig. 9c: are these high values for drizzle drops? They look too large for cloud droplets.

**There is certainly likely to be a contribution from drizzle sized drops to the retrieved values. We have changed the text from "more likely to form drizzle" to "more likely to contain drizzle sized drops"**

9. P15, L26: Is the number of 60 micron for rain drop radius of diameter?

**This is diameter and has been clarified in the revised text.**

10. P17, L18: Panel c of Fig. 13 doesn't seem to be aligned with the other panels.

**Thank you for spotting this. The figure has been corrected.**

11. P17, L32-34: Is there observation showing that the thin/quiescent clouds can last for hours and rain at 10 mm d-1? LES model results seem to show that the active cumulus only last for minutes once they start raining but the resulting cold pool outflow can produce new active clouds in the previous quiescent cloud area.

**The satellite and in-situ observations presented in Wood et al. (2018) do show that these thin quiescent veil clouds can persist for several hours before dissipating. The in-situ observations in that study also show very similar microphysical characteristics to the example presented in this manuscript. It is noted in Wood et al. (2018) that during the temporal evolution of these clouds, the largest drizzle particles are likely to be removed via sedimentation out of cloud base, leaving behind the smaller drops. In this scenario, we would expect the precipitation rate to decrease through the cloud lifetime. We therefore modify the sentence in the manuscript to**

 *"As discussed in Wood et al. (2018)…..mesoscale ascent, or that the smaller drops persist in the thin cloud layer once the largest drizzle particles have fallen out below cloud base, would likely be….."*

12. P18, L31: What's the background level of BC and CO at the surface?

**For the time period where the clean airmass associated with the edge of the POC is over Ascension Island (23 UTC on the 5th Sep to 15 UTC on 6th Sep, see Fig 14 of the submitted manuscript), the mean CO and BC values measured were 70 ppb and 44 ng/m3. The CO is in accordance with the**

**airborne data in the surface mixed layer deep within the POC (see Fig 8b of the submitted manuscript). The aircraft measurements of BC deep in the POC were however lower at approximately 10 ng/m3, although this increased across the POC boundaries (see Fig d of the submitted manuscript). This could simply reflect i) increased removal of BC via more efficient cloud processing and precipitation deep in the POC and ii) lateral mixing of aerosol across the POC boundaries. These values can be compared to the CO and BC measured from the long-term LASIC measurements as reported in Pennypacker et al. (2019). On days that exhibit very clean aerosol conditions during the biomass burning season, the median CO concentration is 69 ppb, with an inter-quartile range of 62 to 74 ppb. For BC, the median value is 51 ng/m3 with an interquartile range of 23 to 120 ng/m3. Outside of the biomass burning season, these values are lower, with median values of 59 ppb (IQR 55 to 65 ppb) and 20 ng/m3 (IQR 12 to 45 ng/m3) respectively. This suggests that there are still often biomass burning signatures in the boundary layer at Ascension Island on days that exhibit low aerosol concentrations during the burning season. In the revision, we have added the following text:**

*The mean CO and BC values measured at the LASIC site between 23 UTC on the 5th September to 15 UTC on 6th September were 70 ppb and 44 ng/m3 . These are in accordance with values reported from the longer term LASIC measurements on days that exhibit very clean aerosol conditions during the biomass burning season, where the median (inter-quartile range) of CO was 69 ppb (62-74 ppb) and BC was 51 ng/m3 (23-120 ng/m3 ) (Pennypacker et al. , 2019).*

13. P19, L18: is "20 to 12%" correct?

**This reflects a decrease in the open cell frequency in the Muhlbauer et al. (2014) study from August through to October 2008. We have changed the word "ranges" to "changes" in this sentence to make that clearer.**

14. P20, L5-9: the manual identification of POC is a little arbitrary. By looking at the examples in Fig. 17, it seems that the POC identification is quite conservative. Does that mean the 0.25 open cell fraction represent a lower bound? Also, in the formulation, were clear-sky and land surface pixels counted in P_ALL or P_BLACK?

**We agree that the identification of open cells is subjective and as we noted in the submitted manuscript, we expect that our method could miss small regions, that might for example be identified in an automated process like that used by Muhlbauer et al. (2014). So yes, it is fair to say that our estimate could be a lower bound and we have amended the text to state that.**

**P_ALL includes both cloudy, clear sky and missing data pixels – it is simply the total number of pixels in the image within the region of interest. P_BLACK represents the missing data pixels only. In reality, these could be clear sky or cloudy pixels, but the formulation removes these missing data regions from the open cell fraction calculation. As we do the calculation over ocean i.e. the blue shaded region in Fig 16, then the calculation does not include any land points, with the exception of small islands that will not impact the results (those pixels would be included in P_ALL and could be in P_RED or P_BLACK if appropriate).**

**References**

Haywood, J. M. et al.: Overview of the CLoud-Aerosol-Radiation Interaction and Forcing: Year-2017 (CLARIFY-2017) measurement campaign, in preparation, 2020.

Lock, A.: Factors influencing cloud area at the capping inversion for shallow cumulus clouds, Quart. J. Roy. Meteorol. Soc., 135, 941-952, https://doi.org/10.1002/qj.424, 2009.

Stevens, B., Ackerman, A. S., Albrecht, B. A., Brown, A. R., Chlond, A., Cuxart, J., Duynkerke, P. G., Lewellen, D. C., MacVean, M. K., Neggers, R. J. A., Sanchez, E., Siebesma, A. P., and Stevens, D. E.: Simulations of trade wind cumulus under a strong inversion, J. Atmos. Sci., 58, 1870-1891, https://doi.org/10.1175/1520-0469(2001)058<1870:SOTWCU>2.0.CO;2, 2001.

Wu, H., et al.: Vertical and temporal variability of the properties of transported biomass burning aerosol over the southeast Atlantic during CLARIFY-2017, in preparation, 2020.